# TetSphere Splatting:
# Representing High-Quality Geometry with Lagrangian Volumetric Meshes

**Minghao Guo**[1*], **Bohan Wang**[1,2*], **Kaiming He**[1], **Wojciech Matusik**[1]
[1]MIT CSAIL, [2]National University of Singapore

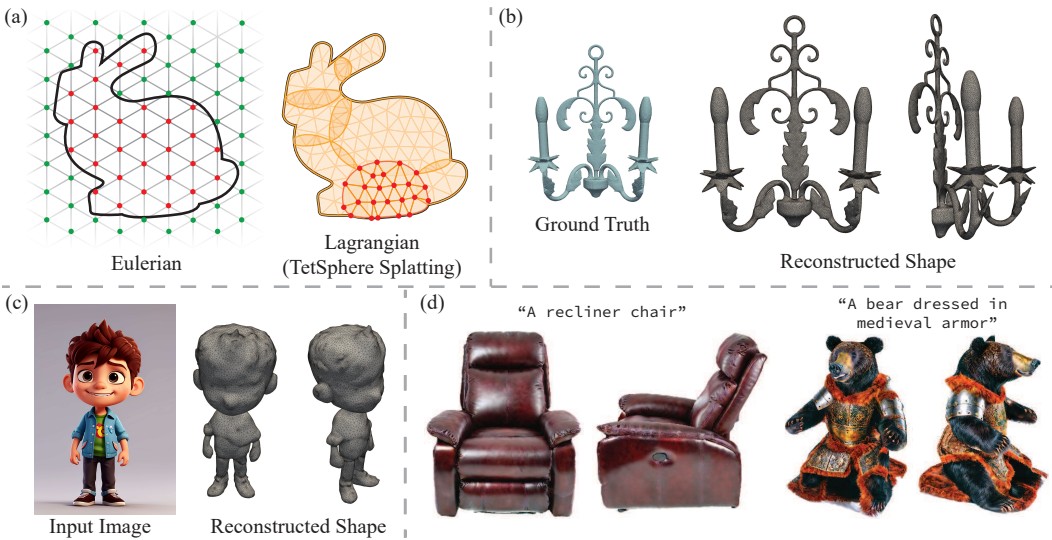

Figure 1: (a) Eulerian vs. Lagrangian geometry representations: Compared to Eulerian methods that rely on a fixed grid, TetSphere splatting, a Lagrangian method, uses a set of volumetric tetrahedral spheres that deform to represent the geometry. TetSphere splatting supports applications such as reconstruction, image-to-3D, and text-to-3D generation (b-d).

## Abstract

We introduce TetSphere Splatting, a Lagrangian geometry representation designed for high-quality 3D shape modeling. TetSphere splatting leverages an under-used yet powerful geometric primitive – volumetric tetrahedral meshes. It represents 3D shapes by deforming a collection of tetrahedral spheres, with geometric regularizations and constraints that effectively resolve common mesh issues such as irregular triangles, non-manifoldness, and floating artifacts. Experimental results on multi-view and single-view reconstruction highlight TetSphere splatting's superior mesh quality while maintaining competitive reconstruction accuracy compared to state-of-the-art methods. Additionally, TetSphere splatting demonstrates versatility by seamlessly integrating into generative modeling tasks, such as image-to-3D and text-to-3D generation. Code is available at `https://github.com/gmh14/tssplat`.

## 1 Introduction

Accurate 3D shape modeling is critical for many real-world applications. Recent advancements in reconstruction (Mildenhall et al., 2020; Wang et al., 2021b; Kerbl et al., 2023), generative modeling (Poole et al., 2022; Liu et al., 2023b;c; Long et al., 2023), and inverse rendering (Mehta et al., 2022; Nicolet et al., 2021; Palfinger, 2022) have significantly improved the geometric precision and visual quality of 3D shapes, pushing the boundaries of automatic digital asset generation.

---
[*]Both authors contributed equally to this research.

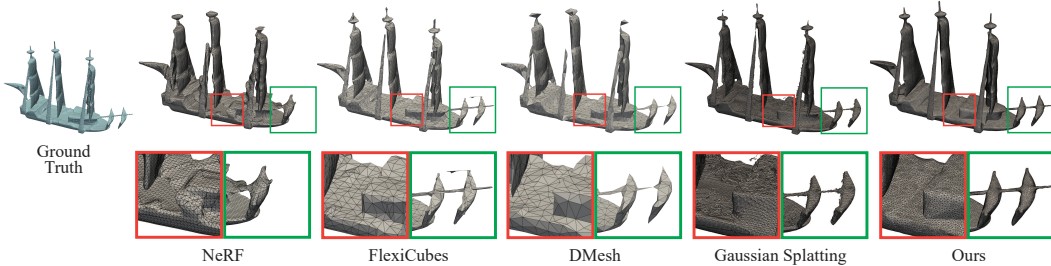

Figure 2: Visual comparison of mesh quality across widely used shape representations, including NeRF (Mildenhall et al., 2020), FlexiCubes (Shen et al., 2023b) (Eulerian), DMesh (Son et al., 2024), and Gaussian Splatting (Huang et al., 2024) (Lagrangian). These methods exhibit mesh quality issues, such as irregular or degenerated triangles, non-manifoldness, and floating artifacts. Our method demonstrates uniform surface triangles, improved mesh quality, and structure integrity.

Central to these advancements are geometry representations, which can be broadly categorized into two types: *Eulerian* and *Lagrangian* representations. *Eulerian* representations describe a geometry on a set of pre-defined, fixed coordinates in 3D world space, where each coordinate position is associated with properties, such as occupancy within the volume or distance from the surface. Widely used Eulerian representations include neural networks that take *continuous* spatial coordinates as input to model density fields (Mildenhall et al., 2020) or signed distance functions (Wang et al., 2021b; Mehta et al., 2022), as well as deformable grids that use *discrete* coordinates, with signed distance values defined at grid vertices (Shen et al., 2021; Gao et al., 2022; Shen et al., 2023b). Despite their popularity, Eulerian representations face a trade-off between computational complexity and geometry quality: Capturing intricate geometric details of the shape requires either a high-capacity neural network or a high-resolution grid, both of which are computationally expensive to optimize in terms of time and memory. This trade-off often limits Eulerian representations when modeling thin, slender structures, as their pre-defined resolution is often insufficient to capture fine details.

Recently, there has been a growing shift in the community towards *Lagrangian* representations, which are typically more computationally efficient than Eulerian methods (Kerbl et al., 2023; Guédon & Lepetit, 2024; Huang et al., 2024; Son et al., 2024; Chen et al., 2024). Lagrangian representations describe a 3D shape by tracking the movement of a set of geometry primitives in 3D world space. These geometry primitives can be adaptively positioned based on the local geometry of the shape, typically requiring fewer computational resources than Eulerian methods, especially when modeling shapes with fine geometric details. An illustrative comparison between Eulerian and Lagrangian geometry representation is shown in Fig. 1 (a). Two of the most commonly used Lagrangian primitives are 3D Gaussians (Kerbl et al., 2023), which represent geometry using point clouds, and surface triangles (Son et al., 2024; Chen et al., 2024), which are coupled with their connectivity to form surface meshes. While these Lagrangian representations are favored for their computational efficiency, they often struggle with poor *mesh quality* due to their reliance on tracking individual points or triangles, which can lack overall structural coherence. For instance, Gaussian point clouds can move freely in space, often resulting in noisy meshes, while surface triangles, when coupled with connectivity, can form non-manifold surfaces or irregular, and sometimes degenerated, triangles. The resulting geometry, exhibiting these geometric issues, is unsuitable for downstream tasks such as rendering and simulation, where high-quality meshes are crucial for high fidelity.

To address these challenges, we propose a novel Lagrangian geometry representation, *TetSphere Splatting*, designed to construct geometry with an emphasis on producing high-quality meshes. Our key insights stem from the fact that existing Lagrangian primitives are too fine-grained to ensure high-quality meshes. Mesh quality depends not only on individual primitives but also on their interactions: For example, the absence of irregular or degenerated triangles relies on the proper alignment of primitives, while manifoldness depends on how well they are connected. Our representation uses volumetric tetrahedral spheres, termed *TetSphere*, as geometric primitives. Unlike existing primitives that are individual points or triangles, each TetSphere is a volumetric sphere composed of a set of points connected through tetrahedralization. Initialized as a uniform sphere, each TetSphere can be deformed into complex shapes. Together, a collection of these deformed TetSpheres represents a 3D shape, which is in line with the Lagrangian approach. This more structured primitive allows geometric regularization and constraints to be imposed among points within each TetSphere, ensuring

mesh quality is maintained throughout deformation. The volumetric nature of TetSpheres also establishes a cohesive arrangement of points throughout the volume, ensuring structural integrity and effectively reducing common surface mesh issues such as irregular triangles or non-manifoldness.

We further present a computational framework for TetSphere splatting. Similar to Gaussian splatting (Kerbl et al., 2023), our method "splats" TetSpheres to conform to the target shape. We formulate the deformation of TetSphere as a geometric energy optimization problem, consisting of differentiable rendering loss, bi-harmonic energy of the deformation gradient field, and non-inversion constraints, all effectively solvable via gradient descent. For evaluation, we conduct quantitative comparisons on two tasks to assess the geometry quality: multi-view and single-view reconstruction. In addition to the commonly used metrics for evaluating reconstruction accuracy, we introduce three metrics to evaluate mesh quality, focusing on key aspects of 3D model usability: surface triangles uniformity, manifoldness, and structural integrity. Compared to state-of-the-art methods, TetSphere splatting demonstrates superior mesh quality while maintaining competitive performance on other metrics. Furthermore, as demonstrations of its versatility, we showcase TetSphere splatting's utility in downstream applications of 3D shape generation from both single images and text.

## 2 RELATED WORK

**Eulerian and Lagrangian geometry representations.** The differentiation between Eulerian and Lagrangian representations originates from computational fluid dynamics (Chung, 2002) but extends more broadly into computational geometry and physics. Using fluid simulation as an analogy, an Eulerian view would analyze fluid presence at fixed points in space, whereas a Lagrangian perspective follows specific fluid particles. Neural implicit representations, such as DeepSDF (Park et al., 2019) and NeRF (Mildenhall et al., 2021) are modern adaptations of Eulerian concepts, processing 3D positions as inputs to neural networks. These methods theoretically allow for infinite resolution through NN-based parameterization but can result in slow optimization speeds due to NN training. Explicit or hybrid Eulerian representations, such as DMTet (Shen et al., 2021) and TetGAN (Yang et al., 2019), incorporate explicit irregular grids but can still cause substantial memory usage for high-resolution shapes. Mosaic SDF (Yariv et al., 2024) uses Lagrangian volumetric grids that move in space but are designed for 3D generation tasks only where ground-truth shapes are required. Gaussian splatting (Tang et al., 2023a) exemplifies a Lagrangian approach by moving Gaussian point clouds in space. Surface triangles (Son et al., 2024; Chen et al., 2024) are another example of a Lagrangian approach, where the surface is discretized into a collection of connected triangular elements tracked individually. Our TetSphere can be viewed as introducing constraints among points due to tetrahedral meshing, with enhanced mesh quality.

**3D object reconstruction.** 3D reconstruction is an inherently ill-posed problem, and extensive research has been dedicated to addressing it (Fu et al., 2021; Fahim et al., 2021). Early approaches utilized a combination of 2D image encoders and 3D decoders trained on 3D data with both explicit representations, including voxels (Chen & Zhang, 2019; Xie et al., 2019; 2020), meshes (Wang et al., 2018; Gkioxari et al., 2019), and point clouds (Mandikal et al., 2018; Groueix et al., 2018), and implicit representations such as NeRF (Yu et al., 2020; Jang & Agapito, 2021; Müller et al., 2022), SDF(Park et al., 2019; Mittal et al., 2022; Xu et al., 2019), and occupancy networks(Mescheder et al., 2019; Bian et al., 2021). Recently, an active research direction has been leveraging 2D generative models for 3D reconstruction, including the use of SDS and supplementary losses (Lin et al., 2023; Sun et al., 2023; Liu et al., 2023b; Melas-Kyriazi et al., 2023; Shen et al., 2023a; Xu et al., 2022; Gu et al., 2023; Deng et al., 2022). The recent introduction of large-scale 3D dataset propelled feed-forward large reconstruction models (Hong et al., 2023b; Wang et al., 2023b; Xu et al., 2023; Weng et al., 2024; Tang et al., 2024; He & Wang, 2023; Tochilkin et al., 2024; Wei et al., 2024; Zhang et al., 2024; Xu et al., 2024b). Their feed-forward inference accelerates the speed of 3D reconstruction, but often at a sacrifice of relatively low resolution and geometry quality. There is also a body of work from the inverse rendering community focused on 3D reconstruction (Nicolet et al., 2021; Palfinger, 2022), including two leveraging explicit Lagrangian representations with differentiable rendering losses (Nicolet et al., 2021; Palfinger, 2022; Vicini et al., 2022; Worchel et al., 2022). Nicolet et al. (2021) uses Laplacian preconditioning when computing the gradient. Palfinger (2022) introduces an adaptive remeshing algorithm based on estimating the optimal local edge length. Both methods use a single surface sphere, whereas our method uses multiple tetrahedral spheres as primitives. Detailed discussion and experimental comparisons are provided in Appendix A.

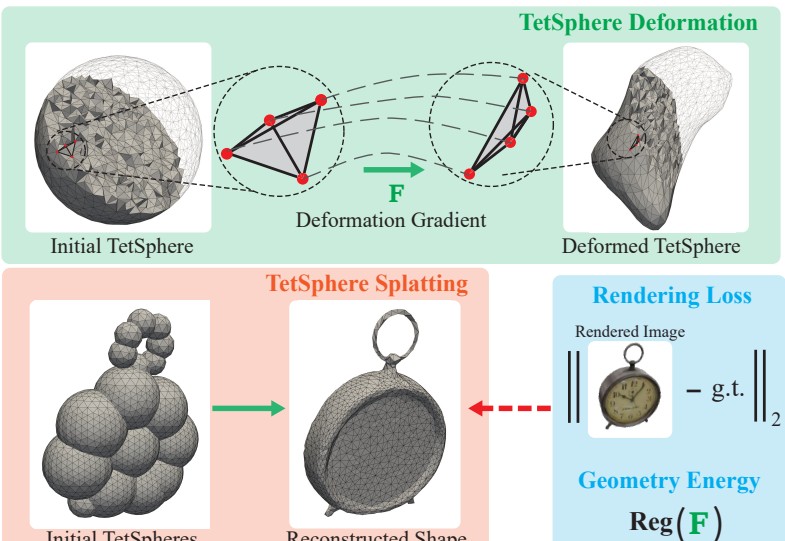

Figure 3: Overall pipeline: TetSphere splatting represents a 3D shape using a collection of Tet-Spheres. Each TetSphere is a tetrahedral sphere that can be deformed from its initial uniform state through deformation gradient. The deformation process is optimized by minimizing rendering loss and geometric energy terms.

Our method also relates to text-to-3D content generation as it is one of the applications of TetSphere splatting. We leave a detailed discussion on these related works in Appendix J.

## 3 TETSPHERE SPLATTING

We use tetrahedral spheres as our primitive of choice. Unlike point clouds, tetrahedral meshes enforce structured local connectivity between points owing to tetrahedralization. This preserves the geometric integrity of the 3D shape and also enhances the surface quality by imposing geometric regularization across the entire mesh interior. We formulate the reconstruction of shapes through Tet-Sphere splatting as a deformation of tetrahedron spheres. Starting from a set of tetrahedral spheres, we adjust the positions of their vertices to align the rendered images of these meshes with the corresponding target multi-view images. Vertex movement is constrained by two geometric regularizations on the tetrahedral meshes, derived from the field of geometry processing (Bærentzen et al., 2012). These regularizations, which penalize the non-smooth deformation (via bi-harmonic energy) and prevent the inversion of mesh elements (via local injectivity), have proven effective in ensuring that the resulting tetrahedral meshes are of superior quality and maintain structural integrity. Fig. 3 illustrates the overall pipeline.

### 3.1 TETRAHEDRAL SPHERE PRIMITIVE

The primitive of TetSphere splatting is a tetrahedralized sphere, called *TetSphere*, with $N$ vertices and $T$ tetrahedra. By applying principles from the Finite Element Method (FEM) (Sifakis & Barbic, 2012), the mesh of each sphere is composed of tetrahedral elements, with each tetrahedron constituting a 3D discrete piecewise linear volumetric entity. We denote the position vector of all vertices of the $i$-th deformed sphere mesh as $x_i \in \mathbb{R}^{3N}$. The deformation gradient of the $j$-th tetrahedron in the $i$-th sphere is denoted as $\mathbf{F}_{\mathbf{x}}^{(i,j)} \in \mathbb{R}^{3\times 3}$, which quantitatively describes how each tetrahedron's shape transforms (Sifakis & Barbic, 2012). Essentially, the deformation gradient $\mathbf{F}_{\mathbf{x}}^{(i,j)}$ serves as a measure of the spatial changes a tetrahedron undergoes from its original configuration to its deformed state. Refer to Fig. 3 for a visual explanation and Appendix F for an in-depth derivation.

Rather than using a single sphere, our representation employs a collection of spheres to accurately represent arbitrary shapes. Consequently, the complete shape is the union of all spheres. By adopting multiple spheres, this approach ensures that each local region of a shape is detailed independently, enabling a highly accurate representation. Moreover, it allows for the representation of shapes with

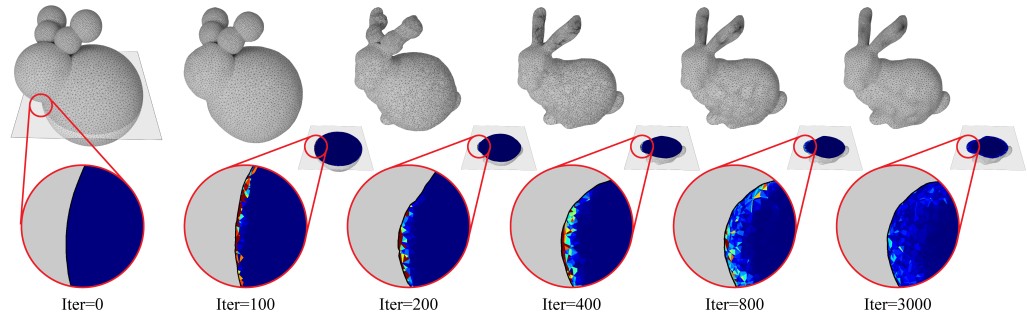

Figure 4: TetSphere splatting with deforming tetrahedral spheres. Color-coded regions represent the bi-harmonic energy values (red: high, blue: low) across tetrahedra, one of the geometric regularizations employed in our deformation optimization process.

arbitrary topologies. Such a claim is theoretically guaranteed by the paracompactness property of manifold shapes (James, 2000).

Using tetrahedral spheres offers several technical benefits compared with prevalent representations for object reconstruction, as demonstrated in Fig. 2:

- Compared to neural representations (e.g., NeRF), our tetrahedral representation does not rely on neural networks, thus inherently accelerating the optimization process.
- Compared to Eulerian representations (such as DMTet), our approach entirely avoids the need for iso-surface extraction – an operation that often degrades mesh quality owing to the predetermined resolution of the grid space.
- Compared to other Lagrangian representations, such as Gaussian point clouds and triangle meshes, our method offers a volumetric representation through the use of tetrahedral meshes. Each tetrahedron imposes constraints among vertices, leading to superior mesh quality.

### 3.2 TETSPHERE SPLATTING AS SHAPE DEFORMATION

To reconstruct the geometry of the target object, we deform the initial TetSpheres by changing their vertex positions. Two primary goals govern this process: ensuring the deformed TetSpheres align with the input multi-view images and maintaining high mesh quality that adheres to necessary geometry constraints. Fig. 4 illustrates the iterative process of TetSphere splatting.

To maintain the mesh quality, we leverage bi-harmonic energy – defined in the literature on geometry processing (Botsch & Sorkine, 2007) as an energy quantifying smoothness throughout a field – to the deformation gradient field. This geometric regularization ensures the smoothness of the deformation gradient field across the deformation process, thus preventing irregular mesh or bumpy surfaces. It is important to highlight that this bi-harmonic regularization does *not* lead to over-smoothness of the final result. This is because the energy targets the deformation gradient field, which measures the *relative* changes in vertex positions, rather than the *absolute* positions themselves. Such an approach allows for the preservation of sharp local geometric details, akin to techniques used in physical simulations (Wen et al., 2023). Furthermore, we introduce a geometric constraint to guarantee local injectivity in all deformed elements (Schüller et al., 2013). This ensures that the elements maintain their orientation during the deformation, avoiding inversions or inside-out configurations. This constraint can be mathematically expressed as $\det(\mathbf{F}_{\mathbf{x}}^{(i,j)}) > 0$. Importantly, these two terms – bi-harmonic energy for smoothness and local injectivity for element orientation – are universally applicable to any tetrahedral meshes, stemming from their fundamental basis in geometry processing (Bærentzen et al., 2012). More details are discussed in Appendix L.

Let $\mathbf{x} = [x_1, ..., x_M] \in \mathbb{R}^{3NM}$ denote the positions of vertices across all $M$ TetSpheres, and $\mathbf{F}_{\mathbf{x}} \in \mathbb{R}^{9MT} = [\text{vec}(\mathbf{F}_{\mathbf{x}}^{(1,1)}), ..., \text{vec}(\mathbf{F}_{\mathbf{x}}^{(M,T)})]$ denote the flattened deformation gradient fields of all TetSpheres, where $N$ and $T$ denote the number of vertices and tetrahedra within each TetSphere, respectively. In the bi-harmonic energy, the Laplacian matrix is defined based on the connectivity of the tetrahedron faces, denoted as $\mathbf{L} \in \mathbb{R}^{9MT \times 9MT}$. This matrix is block symmetric, where

each block $\mathbf{L}_{pq} \in \mathbb{R}^{9 \times 9}, p \neq q$ is set to a negative identity matrix $-I$ if the $p$-th and $q$-th tetrahedron shares a common triangle; or $kI$ for $\mathbf{L}_{pp}$, where $k$ is the number of neighbors of the $p$-th tetrahedron. The deformation of the TetSpheres is formulated as an optimization problem:

$$\min_{\mathbf{x}} \ \mathbf{\Phi}(R(\mathbf{x})) + ||\mathbf{L}\mathbf{F}_{\mathbf{x}}||_2^2$$
$$\text{s.t. } \det(\mathbf{F}_{\mathbf{x}}^{(i,j)}) > 0, \ \forall i \in \{1, ..., M\}, \ j \in \{1, ..., T\}, \tag{1}$$

where $R(\cdot)$ is the rendering function, $\mathbf{\Phi}(\cdot)$ is the rendering loss matching the union of deformed tetrahedral spheres with the input images. The second term regulates the bi-harmonic energy across the deformation gradient field. The non-inversion constraint ensures that tetrahedra maintain their orientation. To manage this constrained optimization, we reformulate it by incorporating the non-inversion hard constraint as a soft penalty term into the objective,

$$\min_{\mathbf{x}} \ \mathbf{\Phi}(R(\mathbf{x})) + w_1||\mathbf{L}\mathbf{F}_{\mathbf{x}}||_2^2 + w_2 \sum_{i,j}(\min\{0, \det(\mathbf{F}_{\mathbf{x}}^{(i,j)})\})^2, \tag{2}$$

allowing for optimization via standard gradient descent solvers.

In the proposed optimization framework, three considerations have been outlined. 1) The adaptive loss function $\mathbf{\Phi}(\cdot)$, designed to be flexible, supports a variety of metrics, including $l_1$ for color images, MSE for depth images, and cosine embedding loss for normal images. These losses can all be optimized using gradients provided by differentiable renderers, such as mesh rasterizers (Laine et al., 2020).. 2) Given that the tetrahedron is a linear element, the deformation gradient $\mathbf{F}_{\mathbf{x}}^{(i,j)}$ is a linear function of $\mathbf{x}$, making the bi-harmonic energy a quadratic term. 3) The weights $w_1$ and $w_2$ are dynamically adjusted using a cosine scheduler. We provide details of the scheduler's hyperparameters in Appendix I.

## 4 TETSPHERE INITIALIZATION AND TEXTURE OPTIMIZATION

**TetSphere initialization.** Given multi-view images as inputs, we select feature points to initialize the 3D center positions of the TetSpheres. We aim to achieve a uniform distribution of these TetSpheres, ensuring comprehensive coverage of the silhouette depicted in the multi-view images.

We introduce an algorithm, silhouette coverage, inspired by Coverage Axis (Dou et al., 2022) to select initial centers of TetSpheres for an arbitrary shape automatically. This process begins with constructing a coarse voxel grid and initially assigning a zero value to each voxel. By projecting these voxels into the image spaces using the same camera poses as the input multi-view images, voxels within the foreground of all images are marked with a value of 1. These voxel positions are identified as candidate positions of TetSphere centers. From these marked positions, the objective is to pick a minimal subset of candidates to ensure that all candidates are fully encapsulated by Tet-Spheres and centered on these points. This involves placing uniform spheres with varying radius values at all candidate points and choosing a minimal subset that collectively covers all the candidate points. We formulate a linear programming problem to perform the selection efficiently. The detailed formulation is provided in Appendix G. In our implementation, with a voxel grid resolution $300 \times 300$ and $M = 20$, the whole TetSphere initialization completes in $\sim 1$ minute on average.

**Texture optimization.** While the primary purpose of TetSphere splatting is to represent high-quality geometry, its explicit structure allows textures and materials to be directly applied to the surface vertices and faces of the TetSpheres. A key advantage of TetSphere splatting is that the deformation of tetrahedral spheres preserves the surface topology, enabling the seamless integration of advanced material models, such as Disney's principled BRDF (Burley & Studios, 2012), in physically-based rendering for applications like text-to-3D generation. However, it is important to note that texture optimization is an optional feature of our representation. The primary objective of TetSphere splatting remains the accurate representation of geometry, with textures and materials being secondary enhancements. Further details are provided in Appendix H.

## 5 EXPERIMENTS AND RESULTS

We provide detailed quantitative evaluations on 3D reconstruction from both multi-view and single-view images to demonstrate the effectiveness of TetSphere. Additionally, we showcase TetSphere's

Table 1: Multi-View reconstruction results: Evaluating reconstruction accuracy with Chamfer Distance (Cham.) and Volume IoU, alongside mesh quality metrics: Area-Length Ratio (ALR), Manifoldness Rate (MR), and Connected Component Discrepancy (CC Diff.). For additional results on other metrics, please refer to Table 5 in Appendix E.1.

| Method | Geo. Rep. | Cham. ↓ | Vol. IoU ↑ | ALR ↑ | MR(%) ↑ | CC Diff. ↓ |
|---|---|---|---|---|---|---|
| NIE | Eulerian | 0.0254 | 0.1863 | 0.0273 | 72.3 | 7.0 |
| FlexiCubes | Eulerian | 0.0247 | 0.5887 | 0.0722 | 45.5 | 201.3 |
| NeuS | Eulerian | 0.0192 | 0.6182 | 0.0573 | 72.3 | 8.1 |
| VolSDF | Eulerian | 0.0185 | 0.6423 | 0.0622 | 81.8 | 7.3 |
| 2DGS | Lagrangian | 0.0322 | 0.4923 | 0.0209 | 27.3 | 25.1 |
| DMesh | Lagrangian | **0.0136** | 0.5616 | 0.1193 | 9.09 | 3.75 |
| Ours | Lagrangian | 0.0184 | **0.6844** | **0.6602** | **100** | **0.0** |

versatility through case studies on image-to-3D and text-to-3D shape generation. We further analyze the effect of energy coefficients and offer a comparative study on computational costs, particularly focusing on GPU requirements and memory usage for image-to-3D generation with SDS loss, where existing methods struggle with high computational demands.

## 5.1 IMPLEMENTATION DETAILS

Our implementation of the TetSphere initialization algorithm is developed in C++ and uses the Gurobi linear programming solver. The optimization of geometric energies is implemented using CUDA as a PyTorch extension to enhance computational efficiency. 1) For multi-view reconstruction, we render multi-view RGBA and depth images of the ground truth shape as the input of TetSphere splatting. The optimization objective $\Phi(\cdot)$ includes both rendering loss (MSE on the alpha mask) and depth loss, following Son et al. (2024). 2) For single-view reconstruction, we use Wonder3d (Long et al., 2023) to generate six multi-view images with predefined camera poses. The optimization objective $\Phi(\cdot)$ includes rendering loss $l_1$ norm on tone-mapped color and MSE on the alpha mask) and normal loss (cosine loss on normals), following Munkberg et al. (2022). The implementation details for the two applications are described in Appendix I. To enhance robustness and efficiency, we apply a cosine scheduler to scale the coefficients of the geometry loss, formulated as $\eta = 4^{\sin(\frac{t\pi}{2n})}$, where $t$ denotes the current iteration and $T$ represents the total number of iterations.

## 5.2 BASELINES AND EVALUATION PROTOCOL

**Baselines.** For multi-view reconstruction, we quantitatively compare our method with state-of-the-art geometry representations: Neural Implicit Evolution (NIE) (Mehta et al., 2022), Flexi-Cubes (Shen et al., 2023b), 2DGS (Huang et al., 2024), and DMesh (Son et al., 2024). The first two are Eulerian representations, whereas the latter two are Lagrangian representations. For single-view reconstruction, we quantitatively compare with several state-of-the-art methods: Magic123 (Qian et al., 2023), One-2-3-45 (Liu et al., 2023a), SyncDreamer (Liu et al., 2023c), Wonder3d (Long et al., 2023), Open-LRM (Hong et al., 2023b; He & Wang, 2023), and DreamGaussian (Tang et al., 2023a). DreamGaussian is the only Lagrangian representation, as the field of single-view reconstruction is largely dominated by Eulerian methods. For all baselines, we follow their original implementations and use their publicly available codebase to get the results in this paper.

**Evaluation datasets.** For multi-view reconstruction, we follow Son et al. (2024) and use four closed-surface models from the Thingi32 dataset (Zhou & Jacobson, 2016), four open-surface models from the DeepFashion3D dataset (Heming et al., 2020), and two additional models with both closed and open surfaces from the Objaverse dataset (Deitke et al., 2023). We also add the challenging sorter shape from Google Scanned Objects (GSO) dataset (Downs et al., 2022) as it features slender structures. For single-view reconstruction, following prior research (Liu et al., 2023c; Long et al., 2023), we use the GSO dataset for our evaluation, which covers a broad range of everyday objects. The evaluation dataset aligns with those used by SyncDreamer and Wonder3D, featuring 30 diverse objects ranging from household items to animals.

**Metrics.** To assess the accuracy of the reconstruction, we use two commonly used metrics: Chamfer Distance (Cham.) and Volume IoU (Vol. IoU), comparing the ground-truth shapes with the

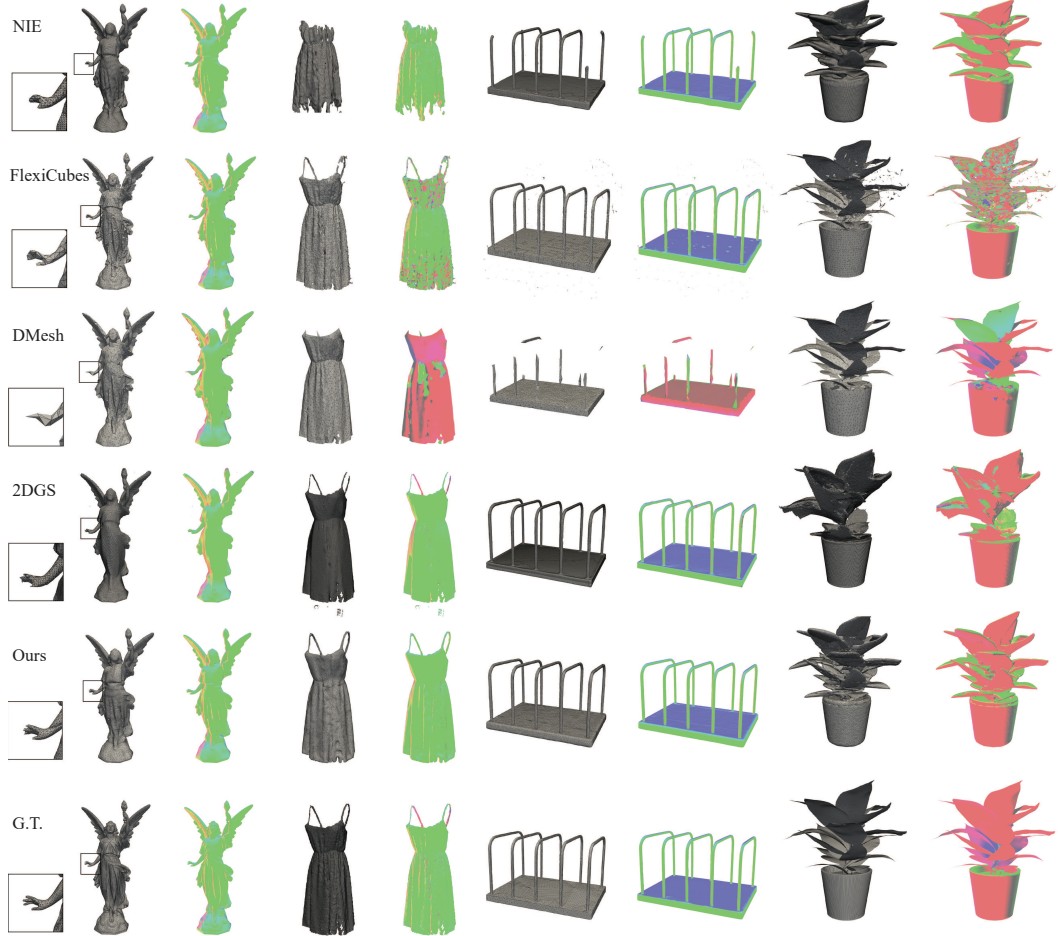

Figure 5: Qualitative results on multi-view reconstruction, with surface mesh visualizations and rendered normal maps. Our method excels over baseline methods regarding mesh quality, less bumpy surface, correct surface orientation, and accurately capturing slender and thin structures.

reconstructed ones. Following established practices, we use the rigid Iterative Closest Point (ICP) algorithm to align the generated shapes with their ground-truth counterparts before metric calculation. For multi-view reconstruction, we also report F-Score, Normal Consistency, Edge Chamfer Distance, and Edge F-Score following prior research (Shen et al., 2023b; Son et al., 2024).

While these metrics effectively measure the reconstruction accuracy, they fail to evaluate mesh quality. We use three additional metrics to assess the geometry quality of the reconstructed shapes: 1) **Area-length Ratio (ALR):** This metric computes the average ratio of a triangle's area to its perime-

Table 2: Single-View reconstruction results on the GSO Dataset: Evaluating reconstruction accuracy with Chamfer Distance (Cham.) and Volume IoU, alongside mesh quality metrics: Area-Length Ratio (ALR), Manifoldness Rate (MR), and Connected Component Discrepancy (CC Diff.).

| Method | Geo. Rep. | Cham. ↓ | Vol. IoU ↑ | ALR ↑ | MR(%) ↑ | CC Diff. ↓ |
|---|---|---|---|---|---|---|
| Magic123 | Eulerian | 0.0516 | 0.4528 | 0.0383 | 100 | 13.7 |
| One-2-3-45 | Eulerian | 0.0629 | 0.4086 | 0.0574 | 96 | 0.83 |
| SyncDreamer | Eulerian | **0.0261** | 0.5421 | 0.0201 | 10 | 0.3 |
| Wonder3d | Eulerian | 0.0329 | 0.5768 | 0.0281 | 100 | **0.0** |
| Open-LRM | Eulerian | 0.0285 | 0.5945 | 0.0252 | 100 | **0.0** |
| DreamGaussian | Lagrangian | 0.0641 | 0.3476 | 0.0812 | 100 | 237.4 |
| Ours | Lagrangian | 0.0351 | **0.6317** | **0.3665** | 100 | **0.0** |

Table 3: Comparison of memory cost and run-time speed on image-to-3D generation with SDS loss. We report the maximal batch size of $256 \times 256$ images that can occupy a 40GB A100 and the run-time speed for training with batch size $4$.

| Method | | Maximal Batch Size↑ | Speed↑ (#iter./s) |
|---|---|---|---|
| Eulerian | Make-it-3D | 4 | 1.22 |
| | Magic123 | 4 | 1.03 |
| | SyncDreamer | 48 | 1.8 |
| | DreamCraft3D | 8 | 1.23 |
| | GeoDream | 8 | 1.30 |
| Lagrangian | DreamGaussian | 80 | 4.43 |
| Lagrangian | Ours | **120** | **6.59** |

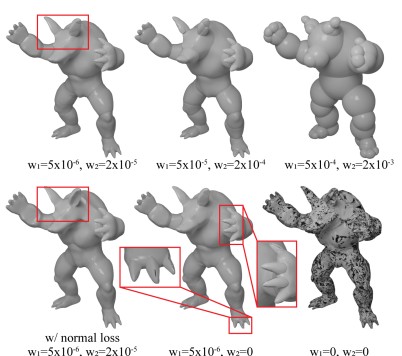

Figure 6: Analysis on geometry energy coefficients. Dark regions indicate flipping of the surface triangles.

ter (scaled by a constant coefficient) within the surface mesh. Values range from $0$ to $1$, where meshes with higher ALR values contain mostly equilateral triangles, thereby indicating superior triangle quality; 2) **Manifoldness Rate (MR):** Manifoldness verifies whether a mesh qualifies as a closed manifold. Non-manifold meshes can manifest anomalies, such as edges shared by more than two faces, vertices connected by edges but not by a surface, isolated vertices and edges, and vertices where more than one distinct surface meets, which can cause problems in downstream applications such as simulation and rendering. We report the percentage of manifold shapes within the evaluation dataset as MR; and 3) **Connected Component Discrepancy (CC Diff.)** from the ground-truth shape: This measure identifies the presence of floating artifacts or structural discontinuities within the mesh, highlighting the integrity and cohesion of the reconstructed shape.

### 5.3 RESULTS

**Multi-view reconstruction.** Table 1 and Table 5 in Appendix E.1 present the quantitative comparison results. Fig. 5 shows the qualitative comparison on four different shapes. Our method excels over baseline methods regarding mesh quality while achieving similar reconstruction accuracy. Specifically, our approach features uniform meshing (as indicated by ALR) and eliminates floating artifacts (as shown by CC Diff.). This improvement is attributed to the volumetric representation of TetSphere and its geometric energy regularization. The qualitative results further support these findings: our method accurately captures slender and thin structures, such as the fingers of Lucy's statue, the sorter, and the dress, which other methods struggle with. Additionally, our method demonstrates correct surface orientation, whereas baseline methods, particularly FlexiCubes, and DMesh, often exhibit inverted triangles, as seen in the noisy rendered normal images. However, our TetSphere approach is less effective when modeling thin shells, such as the leaves of the plant, where DMesh shows better accuracy.

**Single-view reconstruction.** Table 2 shows the comparison results. Fig. 11 in Appendix E.2 shows qualitative results of our method. Our TetSphere splatting excels over baseline methods in terms of mesh quality while also achieving competitive levels of reconstruction accuracy. This improvement is attributed to the regularizations embedded within the TetSphere splatting optimization, ensuring uniform meshing and eliminating common artifacts like floating components. These findings are consistent with the results observed in multi-view reconstruction.

### 5.4 ANALYSIS

**Applications to generative modeling.** TetSphere splatting is a versatile approach that can be effectively applied to generative modeling tasks. We showcase two applications: image-to-3D and text-to-3D generation. Fig. 12 in Appendix E.3 illustrate the results on image-to-3D generation. Our approach outperforms Magic123 and DreamCraft3D regarding mesh quality, achieving smoother surfaces for broad regions, such as animal bodies while retaining local sharp details in areas like eyes and noses. Magic123 tends to produce overly smooth surfaces that sometimes deviate from the

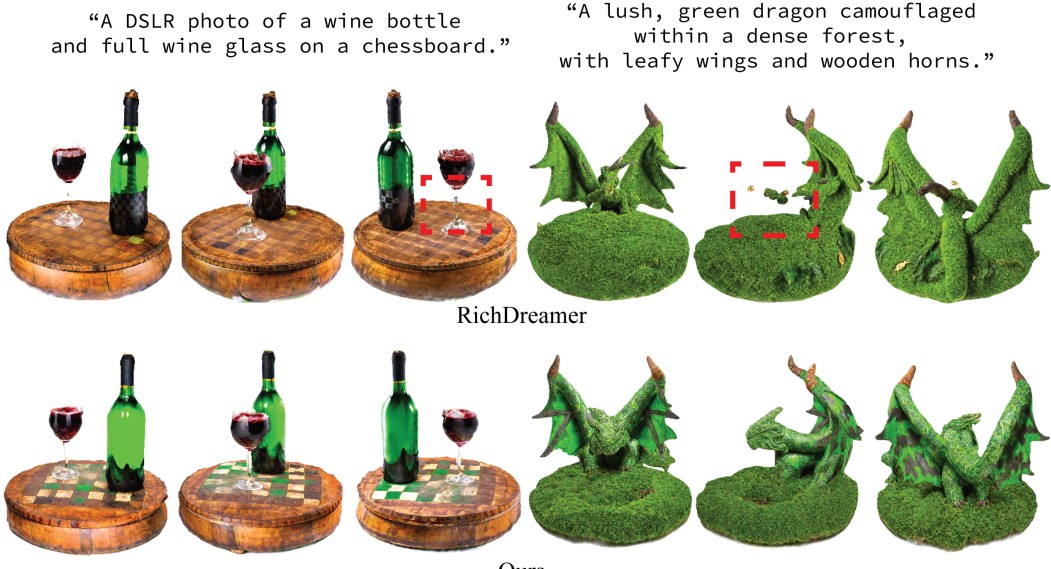

Figure 7: Results on text-to-3D shape generation. Our TetSphere splatting excels in handling slender, thin structures, such as the wine glass stem and the dragon head. More results are available in Appendix E.4.

correct geometry. DreamCraft3D suffers from noisy and bumpy surface meshes, indicating lower mesh quality. Fig. 7 and Fig. 13 in Appendix E.4 shows comparison results on text-to-3D generation. Our TetSphere splatting can produce slender structures such as the dragon's head and the goblet. Furthermore, the results demonstrate that using TetSphere splatting with SDS enhances the geometric detail of the generated shapes, resulting in both diverse and high-quality textures.

**Analysis on computational cost** As shown in Table 5 in Appendix E.1, our method outperforms the Eulerian method NIE regarding running time while remaining competitive with other geometry representation methods. To further assess the performance of TetSphere splatting, we conduct an extreme test by applying it to the task of image-to-3D generation with SDS loss. We evaluate several baseline methods and report the maximum batch size of $256 \times 256$ images that can be processed on a 40GB A100 GPU, as well as the training run-time speed with a batch size of $4$, as shown in Table 3. TetSphere splatting stands out for its minimal memory usage and achieves the fastest run-time speed. This efficiency underscores the benefits of TetSphere's explicit and Lagrangian properties.

**Effects of energy coefficients.** Fig. 6 demonstrates how different energy coefficients influence the reconstruction outcome. Larger coefficients lead to a smoother surface, but excessively small coefficients may cause tetrahedron inversion. In our experiments, we choose $w_1 = 5 \times 10^{-6}, w_2 = 2 \times 10^{-5}$ and apply a cosine increasing schedule.

## 6 CONCLUSION

We introduced TetSphere splatting, a Lagrangian geometry representation for high-quality 3D shape modeling. Our computational framework leverages geometric energy optimization and differentiable rendering to deform TetSpheres into complex shapes, producing results that demonstrate superior mesh quality compared to state-of-the-art methods.

**Limitations.** TetSphere splatting has limitations, including the generation of piecewise meshes rather than a globally consistent tetrahedral mesh and the lack of strong topological guarantees. More details are discussed in Appendix K. Future work could extend TetSphere splatting to leverage direct 3D supervision with volumetric data, eliminating the need for rendering to images. The current TetSphere splatting cannot guarantee topology preservation due to the union of all tetrahedral spheres. This highlights the need for future development in shape generation that can better adhere to topology constraints.

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

Ours

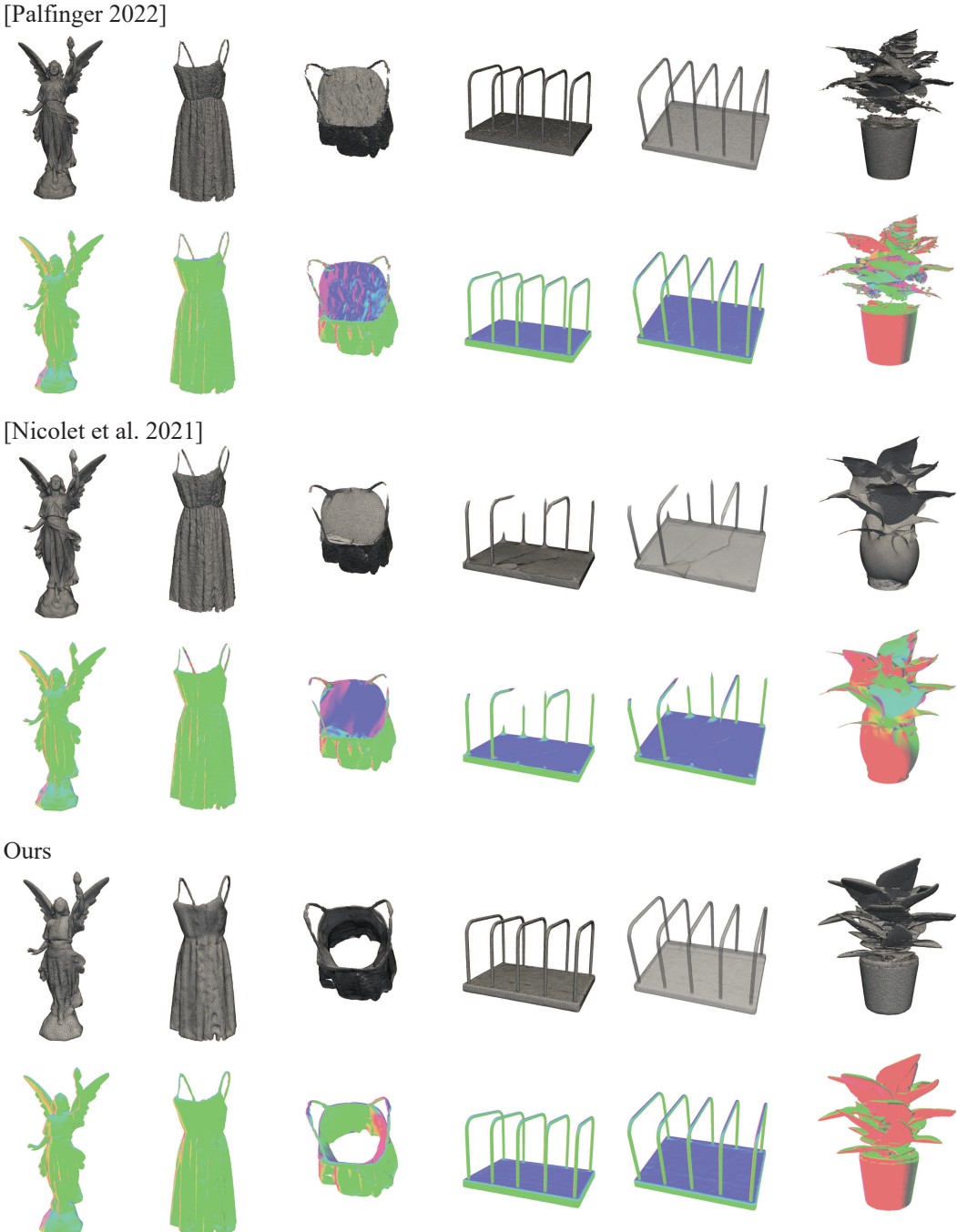

Figure 8: Qualitative comparisons with single-primitive approaches in inverse rendering (Nicolet et al., 2021; Palfinger, 2022).

# A    COMPARISON WITH SINGLE-PRIMITIVE APPROACHES IN INVERSE RENDERING

Our method is related to single-primitive inverse rendering methods, such as Nicolet et al. (2021) and Palfinger (2022). While these existing methods use a single surface sphere for reconstruction, our method leverages a volumetric representation composed of multiple tetrahedral spheres instead of just one. Both Nicolet et al. (2021); Palfinger (2022) and our approach employ regularizations to prevent extreme deformations that can compromise geometric quality, thereby inherently limiting

Table 4: Quantitative comparisons with single-primitive approaches in inverse rendering (Nicolet et al., 2021; Palfinger, 2022). We omit MR and CC Diff., as all three methods achieve identical values.

| Method | Cham. ↓ | Vol. IoU ↑ | ALR ↑ |
|---|---|---|---|
| Nicolet et al. (2021) | 0.0213 | 0.6184 | 0.5235 |
| Palfinger (2022) | 0.0191 | 0.6583 | 0.6332 |
| Ours | **0.0184** | **0.6844** | **0.6602** |

each sphere's expressivity to a certain extent. Nicolet et al. (2021); Palfinger (2022) address this limitation by using intermediate remeshing to accommodate topological changes and manage drastic deformations when the target shape significantly differs from the sphere. However, remeshing results in the reparameterization of the texture, which poses challenges in the context of 3D shape generation pipelines. Maintaining a consistent texture parameterization throughout the optimization process is crucial, as the texture image itself is an optimization variable (see Appendix H). Additionally, remeshing can introduce undesired meshing complications when the surface undergoes topological changes. To prevent topological changes during each iteration, a single primitive must be initialized with a topology that matches the target. By contrast, our method eliminates the need for intermediate remeshing by utilizing multiple TetSpheres alongside stronger regularization terms, including biharmonic energy and non-inversion constraints. By leveraging multiple TetSpheres, each TetSphere undergo a smaller deformations, ensuring robust shape reconstruction.

Table 4 and Fig. 8 illustrate the quantitative and qualitative comparisons of our method against Nicolet et al. (2021) and Palfinger (2022). We used their publicly available codebases and performed multi-view reconstruction on our dataset as described in Sec. 5.2. Our method outperforms these approaches, particularly for shapes with complex topologies. Both Nicolet et al. (2021) and Palfinger (2022) use a single sphere for reconstruction, which limits their ability to handle objects with multiple holes, such as the dress and the sorter. For the dress example (the second and the third columns in Fig. 8), both baseline methods exhibit unrealistic closure of the open areas. In contrast, our approach accurately represents the thin regions and preserves the open areas due to the use of multiple TetSpheres. For the sorter example (the fourth and the fifth columns), the results from Nicolet et al. (2021) exhibit folded, overlapping surfaces with noticeable crumpled regions. This occurs because their deformation regularization is only on the surface and cannot effectively penalize large bending. Our tetrahedral-based method, with its volumetric regularization, mitigates these issues.

# B  ADDITIONAL ABLATION STUDIES

## B.1  ABLATION STUDY ON THE NUMBER OF TETSPHERES

The number of TetSpheres in our method is determined by the initialization algorithm described in Sec. 4 and Appendix G, controlled by the scaling and offset parameters $\alpha$ and $\beta$ that define the initial radius of each TetSphere. Intuitively, larger values of $\alpha$ and $\beta$ result in a sparser distribution of TetSpheres. In this section, we present an ablation study on the reconstruction performance with respect to different numbers of TetSpheres.

We select two examples from the multi-view reconstruction experiment, the dress and the sorter, as these represent the most challenging cases due to their complex topology. For each example, we evaluate five different combinations of $\alpha$ and $\beta$, resulting in $\{1, 59, 87, 323, 434\}$ TetSpheres for the dress and $\{1, 31, 69, 127, 239\}$ for the sorter. Notably, 323 and 127 TetSpheres are the numbers used in our primary method.

The reconstruction results are illustrated in Fig. 9, with corresponding metrics for Chamfer distance, volume IoU, and ALR shown in Fig. 10(a). At one extreme, using only a single TetSphere leads to poor reconstruction, unable to capture the intricate topology of the target shapes. As the number of TetSpheres increases, the reconstruction quality improves significantly. Beyond a certain point, however, such as with 434 and 239 TetSpheres, the quality stabilizes, and the metrics indicate minimal further improvement – although the tessellation density becomes higher.

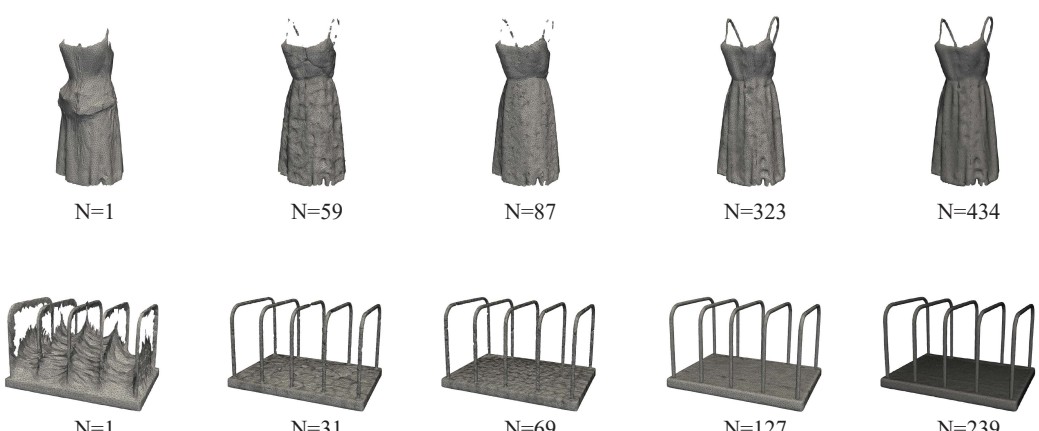

Figure 9: Qualitative reconstruction quality comparison using different numbers of TetSpheres. At one extreme, using only a single TetSphere leads to poor reconstruction, unable to capture the intricate topology of the target shapes. As the number of TetSpheres increases, the reconstruction quality improves. Beyond a certain point, however, the quality stabilizes although the tessellation density becomes higher. Our chosen number of TetSpheres (323 and 127) strikes a balance between achieving high reconstruction quality and maintaining reasonable computational efficiency.

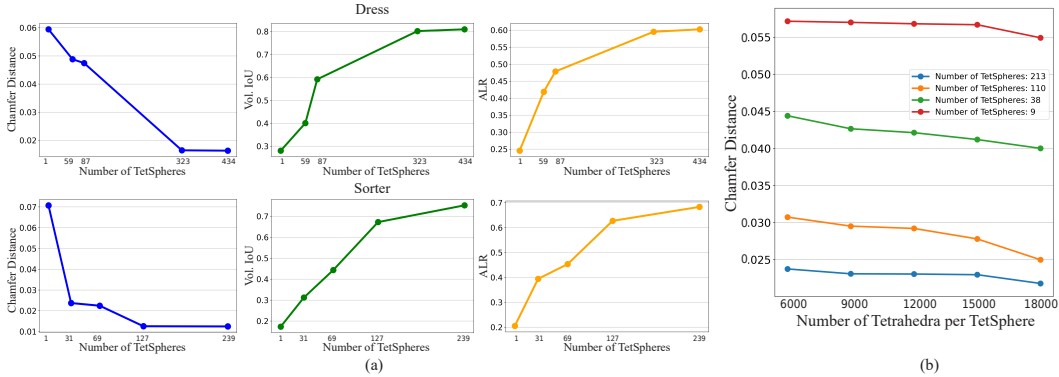

Figure 10: (a) Quantitative comparison using different numbers of TetSpheres. As the number of TetSpheres increases, both the reconstruction accuracy (Chamfer distance and Vol. IoU) and the surface quality (ALR) improve. Beyond 434 and 239, however, the metrics indicate minimal further improvement. (b) Ablation study on the trade-off between the average number of tetrahedra per TetSphere and the total number of TetSpheres. For each setting of the number of TetSpheres, increasing the number of tetrahedra per TetSphere improves reconstruction accuracy. However, the improvements are less pronounced compared to increasing the number of TetSpheres.

An excessive number of TetSpheres increases computational costs due to the additional memory required to store the vertex positions. Our chosen number of TetSpheres strikes a balance between achieving high reconstruction quality and maintaining reasonable computational efficiency.

## B.2 ABLATION STUDY ON THE NUMBER OF TETRAHEDRA PER TETSPHERE

We conducted an ablation study to analyze the trade-off between the average number of tetrahedra per TetSphere and the total number of TetSpheres required to represent the surface. We used four sets of $\alpha$ and $\beta$ values to obtain different granularities of TetSphere counts, averaging $\{9, 38, 110, 213\}$ across all examples in the multi-view reconstruction experiment. For each granularity, we varied the average number of tetrahedra per TetSphere, testing five configurations: $\{6000, 9000, 12000, 15000, 18000\}$. The Chamfer Distance results for these tests are shown in Fig. 10(b).

For each granularity setting, increasing the number of tetrahedra per TetSphere improved reconstruction accuracy. However, the improvements are less significant compared to increasing the number of TetSpheres. While more tetrahedra per TetSphere can provide finer detail within a single TetSphere, the primary determinant of reconstruction accuracy is the overall number of TetSpheres. This is because distributing the surface representation across a larger number of TetSpheres allows for better coverage and adaptability to complex geometric features. Therefore, the number of tetrahedra within each TetSphere plays a secondary role in the overall surface reconstruction quality.

## C    FORMULATIONS OF METRICS

**Area-Length Ratio (ALR):** The ALR is defined as following the definition of TriangleQ in Frey & Borouchaki (1999); Xu et al. (2024a):

$$\text{ALR} = \frac{1}{N} \sum_{i=1}^{N} \frac{6}{\sqrt{3}} \frac{A_i}{P_i h_t},$$

where $N$ is the total number of triangles in the mesh, $A_i$ is the triangle area, $P_i$ is the half-perimeter, and $h_t$ is the longest edge length. This metric ranges from $0$ to $1$, with higher values indicating that the mesh consists primarily of equilateral triangles, thus reflecting superior triangle quality.

**Manifoldness Rate (MR):** The MR checks whether the mesh is a closed manifold. The MR is reported as the percentage of meshes in the evaluation dataset that qualify as closed manifolds:

$$\text{MR} = \frac{\text{Number of manifold meshes}}{\text{Total number of meshes}} \times 100\%.$$

**Connected Component Discrepancy (CC Diff.):** This metric measures the difference in the number of connected components between the reconstructed mesh and the ground-truth shape. Let $C_{\text{rec}}^{(i)}$ and $C_{\text{gt}}^{(i)}$ denote the number of connected components in the reconstructed and ground-truth meshes, respectively. The CC Diff. is defined as:

$$\text{CC Diff.} = \frac{1}{N} \sum_{i=1}^{N} |C_{\text{rec}}^{(i)} - C_{\text{gt}}^{(i)}|,$$

which helps identify floating artifacts or discontinuities, indicating the structural integrity and cohesion of the reconstructed shape.

## D    SIMPLE STUDY ON TETRAHEDRON INVERSION

In the inset figure (left), we provide a 2D illustration showing how inversion can occur, with the inverted region highlighted in red. To further illustrate the importance of our regularization in preventing tetrahedron inversion, we conducted a simple study using a target shape of a sphere that is smaller than the initial Tet-Sphere. As depicted in the inset figure (right), without regularization, the rendering loss alone pushes the surface vertices of the initial Tet-Sphere directly onto the target sphere's surface. This results in a significant number of inverted

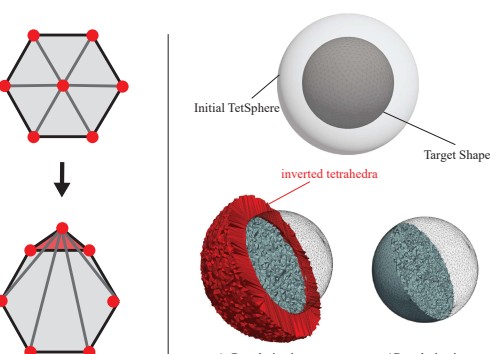

tetrahedra, as highlighted in red. In contrast, with our volumetric regularization in place, the reconstructed shape successfully adheres to the target surface without any inverted tetrahedra, ensuring a high-quality reconstruction.

## E    ADDITIONAL RESULTS

### E.1    MORE QUANTITATIVE RESULTS ON MULTI-VIEW RECONSTRUCTION

Table 5 shows comparisons of multi-view reconstruction on more evaluation metrics.

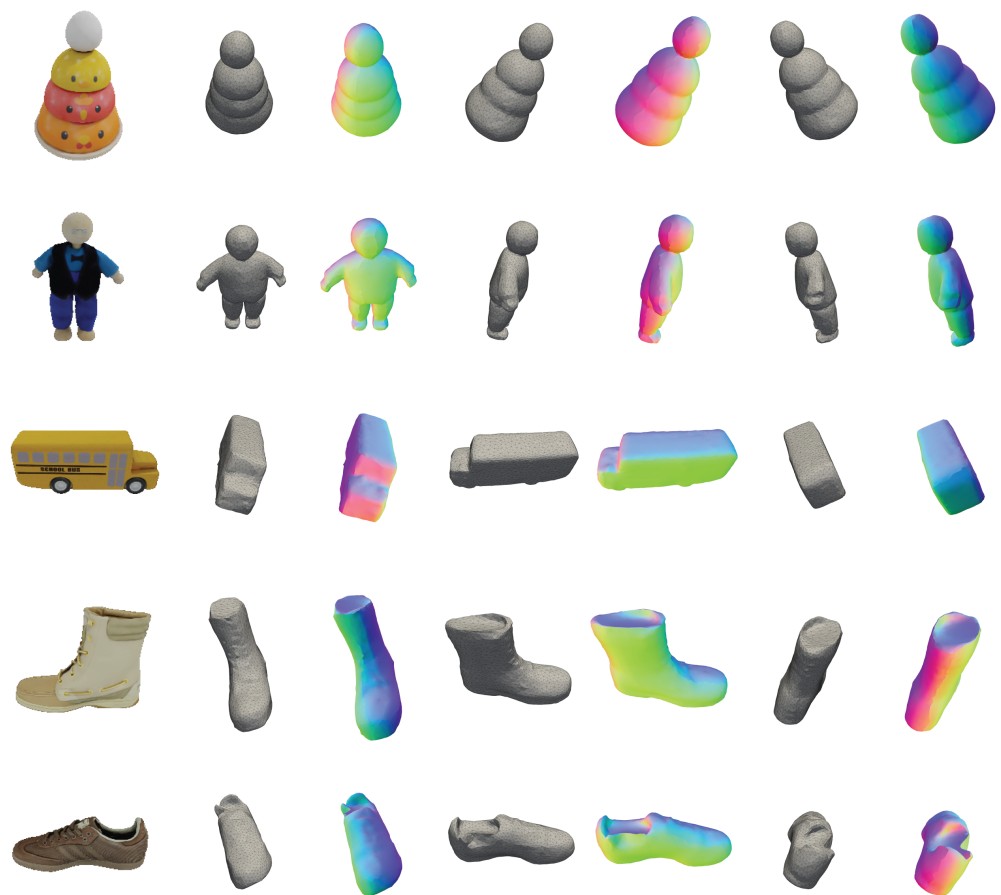

Figure 11: Qualitative results on single-view reconstruction. We use Wonder3d (Long et al., 2023) to generate six multi-view images with predefined camera poses. The optimization objective includes rendering loss $l_1$ norm on tone-mapped color and MSE on the alpha mask) and normal loss (cosine loss on normals), following Munkberg et al. (2022). More details are described in Sec. 5.1

## E.2 QUALITATIVE RESULTS ON SINGLE-VIEW RECONSTRUCTION

Fig. 11 shows the results of single-view reconstruction performed on the GSO dataset. Our Tet-Sphere splatting demonstrates superior mesh quality, effectively capturing sharp geometric features, including the boundaries of the shoes.

## E.3 QUALITATIVE RESULTS ON IMAGE-TO-3D GENERATION

Fig. 12 shows qualitative comparisons of image-to-3D generation with surface mesh visualizations and rendered normal maps. Our generated meshes exhibit minimal noise and high quality, featuring more regular triangle meshing.

Table 5: Additional results on multi-view reconstruction.

| Metric | NIE | FlexiCubes | 2DGS | DMesh | Ours |
|---|---|---|---|---|---|
| F-Score ↑ | 0.486 | 0.502 | 0.632 | 0.605 | **0.653** |
| Normal Consis.↑ | 0.718 | 0.734 | 0.812 | 0.745 | **0.839** |
| Edge Cham. ↓ | 0.039 | 0.034 | 0.015 | 0.049 | **0.014** |
| Edge F-Score ↑ | 0.016 | 0.196 | 0.219 | 0.193 | **0.269** |
| Time (sec.) ↓ | 6436 | **57.67** | 691 | 1434 | 934 |
| # Triangles | 151,223.6 | 30,285.7 | 270,235.5 | 25,767.7 | 70,616.0 |

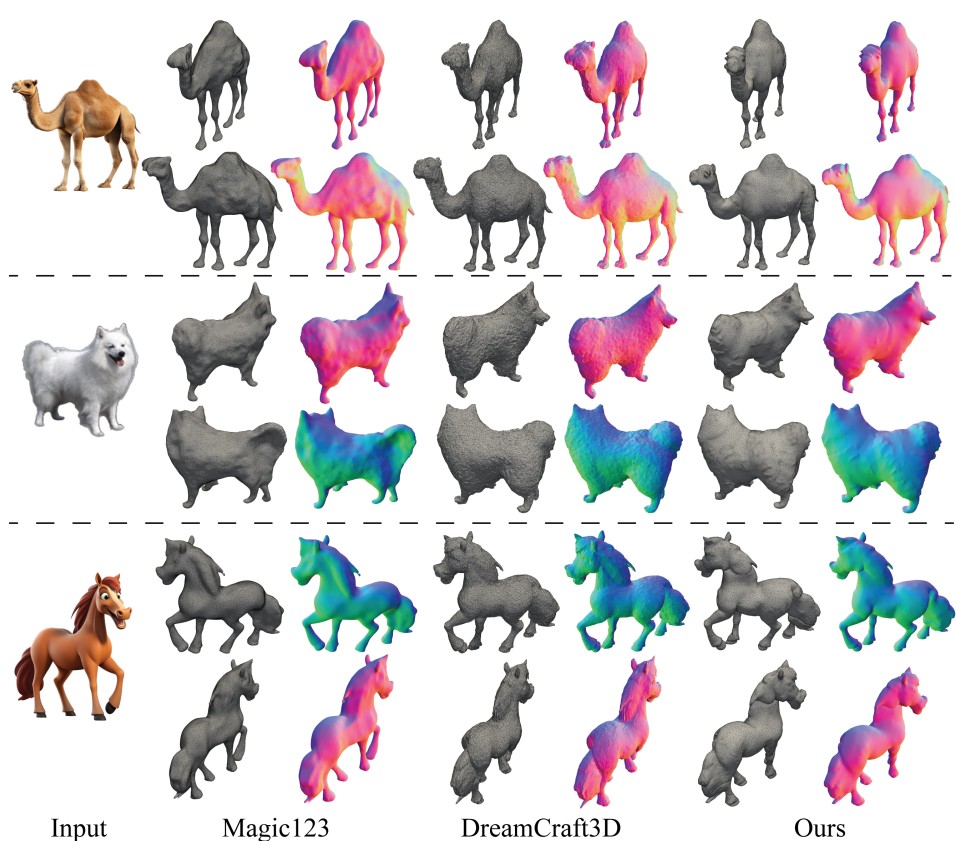

| Input | Magic123 | DreamCraft3D | Ours |

Figure 12: Qualitative comparison on image-to-3D generation with surface mesh visualizations and rendered normal maps. Our generated meshes exhibit less bumpiness and high mesh quality with more regular triangle meshing. We use multi-view images generated using the coarse NeRF fitting stage of DreamCraft3D, sampled on a Fibonacci sphere, and optimize a $2048 \times 2048$ texture image directly using rendering loss. More details are described in Appendix I.

### E.4 MORE RESULTS ON TEXT-TO-3D SHAPE GENERATION

Fig. 13 shows additional results of text-to-3D shape generation. These results highlight our method's capability to construct complicated material, such as reflections, by leveraging the explicit geometry representation.

## F TETRAHEDRON AND ITS DEFORMATION GRADIENT

Following (Sifakis & Barbic, 2012), we treat a tetrahedron as a piecewise linear element. The initial (undeformed) positions of the four vertices of a tetrahedron are denoted by $\mathbf{X} = [\mathbf{X}^{(1)}, \mathbf{X}^{(2)}, \mathbf{X}^{(3)}, \mathbf{X}^{(4)}]$, with each $\mathbf{X}^{(i)} \in \mathbb{R}^3$. Similarly, the positions of the four vertices of a deformed tetrahedron are represented by $\mathbf{x} = [\mathbf{x}^{(1)}, \mathbf{x}^{(2)}, \mathbf{x}^{(3)}, \mathbf{x}^{(4)}]$, where each $\mathbf{x}^{(i)} \in \mathbb{R}^3$. The deformation gradient $\mathbf{F} \in \mathbb{R}^{3 \times 3}$, which quantifies the local deformation of the tetrahedron, is given by:

$$\mathbf{F} = \mathbf{D}_s \mathbf{D}_m^{-1}, \tag{3}$$

$$\mathbf{D}_s := \begin{bmatrix} \mathbf{x}^{(1)} - \mathbf{x}^{(4)} & \mathbf{x}^{(2)} - \mathbf{x}^{(4)} & \mathbf{x}^{(3)} - \mathbf{x}^{(4)} \end{bmatrix}, \tag{4}$$

$$\mathbf{D}_m := \begin{bmatrix} \mathbf{X}^{(1)} - \mathbf{X}^{(4)} & \mathbf{X}^{(2)} - \mathbf{X}^{(4)} & \mathbf{X}^{(3)} - \mathbf{X}^{(4)} \end{bmatrix}. \tag{5}$$

The deformation gradient $\mathbf{F}$ essentially captures how a tetrahedron transforms from its initial state to its deformed state, encompassing both rotation and stretching effects.

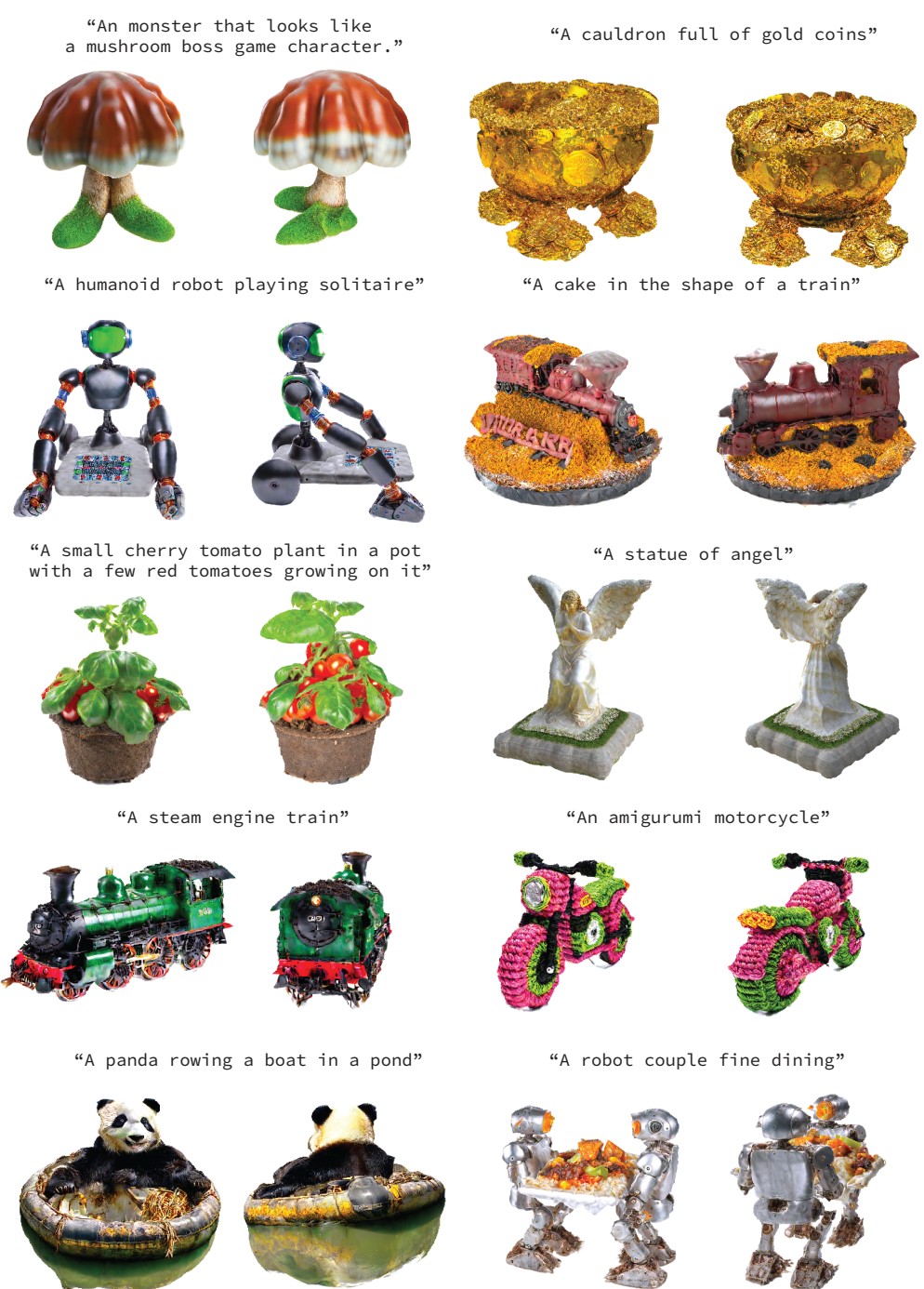

Figure 13: More results on text-to-3D shape generation. TetSphere splatting is optimized using rendering and SDS losses. The SDS loss for geometry is based on the Normal-depth diffusion model in Qiu et al. (2023), and loss for texture uses the albedo diffusion model from the same work. More details are described in Appendix I.

## G FORMULATION OF TETSPHERE INITIALIZATION

Assuming there are a total of $m$ candidate positions obtained from the coarse voxel grid (as shown in Fig. 5), our goal with TetSphere initialization is to select a subset of these candidate points such that the object's shape is adequately covered by tetrahedral spheres centered at these positions. We

first initialize a sphere of fixed radius at each candidate position, where the radius is calculated as $\alpha r + \beta$, where $r$ is the minimum distance from each candidate position to the voxel surface. We use $\alpha = 1.2, \beta = 0.07$ in all our examples. The objective is to select a subset of spheres that collectively cover all voxel positions.

We define a coverage matrix $\mathbf{D} \in \{0, 1\}^{m \times m}$, where each element $d_{ji} \in \{0, 1\}$ indicates whether voxel position $j$ is covered by a sphere centered on candidate position $i$. A binary vector $\mathbf{v} \in \{0, 1\}^{m}$ identifies selected candidate positions, with each element denoting the selection status of corresponding voxel positions. The selection of feature points is formulated as a mixed-integer linear programming problem:

$$\min_{\mathbf{v}} |\mathbf{v}| \quad \text{s.t.} \quad \mathbf{D}\mathbf{v} \geq \mathbb{1}, \tag{6}$$

where $|\cdot|$ is the $l_1$ norm, $\mathbb{1}$ is a vector of ones. This optimization is efficiently solved using standard linear programming solvers.

## H    TEXTURE AND PBR MATERIAL OPTIMIZATION

Material optimization is facilitated by using differentiable rasterizers (Laine et al., 2020), which adjust the textures or materials to closely match the input multi-view color images. A significant advantage of TetSphere splatting is that the deformation of tetrahedral spheres does not alter the surface topology. Unlike methods such as DMTet, which require isosurface extraction and subsequent texture parameterization at each step due to potential changes in the underlying shape, our method necessitates only a single texture parameterization at the beginning of optimization. This parameterization remains consistent throughout the process, significantly enhancing the efficiency of texture optimization.

For scenarios with dense input views, we have found that using textured images as optimization variables is straightforward and yields high-quality results. In cases with sparse input views, we adopt a two-layer MLP that takes the surface vertex positions as inputs and outputs the material parameters, a practice in line with existing methods (Sun et al., 2023; Qiu et al., 2023).

## I    MORE IMPLEMENTATION DETAILS

When a surface mesh is required, we extract the surface triangles from each TetSphere to generate surface spheres, and then perform a union of all these surface spheres to obtain the final surface mesh. While there is no theoretical guarantee of manifoldness in the resulting mesh from union operation, our TetSphere optimization is highly regularized to maintain geometric quality. As a result, we did not encounter non-manifold issues in our experiments.

For image-to-3D shape generation, we obtain multi-view images from the initial stage of Dream-Craft3D (coarse NeRF fitting only), generating 360 views sampled on a Fibonacci sphere (González, 2010). The texture optimization directly employs a $2048 \times 2048$ 2D texture image, circumventing the need for additional neural networks. Here, the optimization objective $\mathbf{\Phi}(\cdot)$ focuses solely on the rendering loss. For text-to-3D generation, we use the initial stage of RichDreamer (Qiu et al., 2023) to obtain multi-view images. We optimize our TetSphere splatting using both the rendering loss and the SDS loss. The SDS loss for geometry is calculated using the Normal-depth diffusion model as described in Qiu et al. (2023). The SDS loss for texture leverages the albedo diffusion model from the same work.

Regarding baselines: For image-to-3D generation, we compare TetSphere splatting with Magic123 and Dreamcraft3D (Sun et al., 2023). Both are multi-stage methods starting with NeRF optimization followed by DMTet to optimize mesh and texture. For text-to-3D, our comparison focuses on Rich-Dreamer, a state-of-the-art method known for incorporating PBR materials and generating shapes, showcasing notable results in 3D generation from text prompts.

## J    RELATED WORK – TEXT-TO-3D CONTENT GENERATION

The recent success of text-to-2D image generation models has spurred a growing interest in generating 3D output from text input. In light of the limited availablity of text-annotated 3D datasets,

methods have been developed that leverage the pre-trained text-to-image models to reason between 2D renderings of 3D models and text descriptions. Early works (Khalid et al., 2022; Jain et al., 2022) adopt the pre-trained CLIP model (Radford et al., 2021) to supervise the generation by aligning the clip text and image embeddings. Recently, 2D diffusion-based generative models (Saharia et al., 2022; Rombach et al., 2022) have powered direct supervision in the image/latent space and achieved superior 3D quality (Poole et al., 2022; Tang et al., 2023b; Chen et al., 2023; Lin et al., 2023; Wang et al., 2023c). Notably, DreamFusion (Poole et al., 2022) introduces the Score Distillation Sampling (SDS) for supervising the NeRF (Mildenhall et al., 2020) optimization using diffusion priors as a score function (i.e., by minimizing the added noise and the predicted noise under the text condition). Follow-up works have since been proposed to improve score sampling formula with Perturb-and-Average and variational method (Wang et al., 2023c;a), improved noise sampling schedules (Huang et al., 2023), various 3D representations (Lin et al., 2023; Tsalicoglou et al., 2023; Chen et al., 2023), text prompts (Armandpour et al., 2023), 3D consistency (Li et al., 2023; Hong et al., 2023a), and prior quality with dedicated diffusion models (Qiu et al., 2023; Shi et al., 2023). Our method leverages the Normal-Depth diffusion model used in (Qiu et al., 2023) for text-to-3D shape generation, which is trained on the large-scale LAION dataset, but replaces the geometry representation with our TetSphere.

## K  LIMITATIONS

While our representation demonstrates improvements in reconstruction quality, there are notable limitations that should be acknowledged. For example, the method only produces piecewise meshes that are overlaid, rather than solving the problem of globally fitting a single tet-mesh. Furthermore, while our regularization helps prevent issues such as tetrahedron inversion and maintains mesh integrity, it does not provide strong topological guarantees. The union operation of multiple TetSpheres can potentially alter the topology and may not fully preserve manifold properties. Addressing these challenges to ensure global mesh consistency and topological guarantees could be valuable directions for future work.

## L  DISCUSSIONS ON BI-HARMONIC ENERGY ON DEFORMATION GRADIENTS

The incorporation of biharmonic energy in our method is inspired by its proven efficacy in geometry processing, as highlighted by prior research (Botsch & Sorkine, 2007). Furthermore, penalizing the non-smoothness of transformations (or deformation gradient) across a domain is a well-explored area that has demonstrated its utility in various geometry processing tasks such as shape deformation (Wang et al., 2021a), deformation transfer (Sumner & Popović, 2004), and mesh alignment Amberg et al. (2007). For the surface regions where observations are available, the surface deformation is primarily influenced by the rendering loss. Conversely, in regions lacking direct observations, our regularization term plays an important role by penalizing high-frequency deformations resulting from non-smooth deformation gradients. This approach ensures reliable results. The biharmonic energy offers stronger regularization compared to first-order harmonic energy, as demonstrated in (Botsch & Sorkine, 2007). Although theoretically higher-order smoothness energies can be considered, in practice, they often lead to numerical instability, as discussed in Jacobson & Panozzo (2017). Therefore, biharmonic energy strikes a balance by providing effective regularization while avoiding numerical issues.

