# OpenReview forum: "TetSphere Splatting: Representing High-Quality Geometry with Lagrangian Volumetric Meshes"
_ICLR.cc/2025/Conference — ICLR 2025 Oral_

### Official Review · Reviewer_fnQo · 2024-10-22

**Soundness:** 3
**Presentation:** 3
**Contribution:** 3
**Rating:** 8
**Confidence:** 4

**Summary:**

This paper uses tetrahedral spheres to represent three-dimensional shapes and constructs its corresponding differentiable inverse rendering process, which improves tasks such as single-view, multi-view reconstruction, and text to 3d shapes.

**Strengths:**

I think it’s a good idea to use tetrahedrons to represent shapes, which is very common in classical geometry processing. I’m excited to see that the authors were able to combine computer vision and geometry processing and make good progress on multiple tasks.

This new method surpasses or catches up with Sota methods on multiple data sets, and because it is based on the tetrahedron geometric primitive, it seems to be able to achieve better topology preservation, manifold preservation and better triangulation quality.

**Weaknesses:**

The implementation details of this paper are not enough for me, and many questions come to my mind when I read this paper.

1. The author does mention the problem of tetrahedron inversion, and cites theories from other papers to show that it is useful. I think it would be easier to understand if there is a simple ablation experiment.
2. The paper does not seem to mention how to extract the surface mesh from the tetrahedral mesh, which I think is also an important part of the whole pipeline. At the same time, how to ensure that there is no non-manifold when extracting the surface is also a problem.
3. How to ensure that the topology is correct? If I understand correctly, this seems to be strongly related to "silhouette coverage", which must be consistent with the topology of GT. I would like to see an analysis of the Euler characteristic (Genus) in the tables.
4. This paper introduces many metrics to evaluate the quality of the mesh, but no corresponding references or mathematical definitions are given, like Manifoldness Rate and Connected Component Discrepancy. I hope to see their mathematical definitions in the appendix, which will make it easier for readers to understand.
5. Regarding mesh quality, first of all, do all compared methods have similar resolution/triangle numbers? If mesh quality is to be compared, giving the triangle numbers is a must. It would be great if the authors could provide a metric for triangle quality, such as the Edge Chamfer Distance in NMC [Chen and Zhang. 2021] and TriangleQ in CWF [Xu et al. 2024].
6. In addition, since your initialization is multiple tetrahedralized spheres, what is the connection between the spheres? Are they multiple independent connected branches? If so, how to deal with the intersection between them, and how to ensure that the final mesh is an independent and connected one?
7. Citation problem, I found multiple citations of same paper in several places, such as One-2-3-45 line 648-653, Syncdreamer line 657-662 etc.

**Questions:**

My main doubts lie in the topological and manifold guarantees claimed by the authors, and I hope to see more detailed analyses and experiments to prove their method. I am also curious about the mesh extraction method and the number and quality of triangles. And if the code is attached, it will also increase the credibility of this paper.

Overall, I like this idea, but it seems to have a lot of minor issues. If the authors can address these issues during rebuttal, I will increase my rating.

---

> ### Author Response · Authors · 2024-11-19
> **Rebuttal**
>
> **W1: Simple study on tetrahedron inversion**
>
> In Appendix D of the revised version, we provide a 2D illustration showing how inversion can occur and include a simple study in 3D demonstrating tetrahedron inversion and how our regularization alleviates this issue. In this study, we fit a target shape of a sphere that is smaller than the initial TetSphere. As shown in the figure in Appendix D, without regularization, the rendering loss alone pushes the surface vertices of the initial TetSphere directly onto the target sphere's surface, resulting in a significant number of inverted tetrahedra. In contrast, our volumetric regularization prevents this issue and avoids tetrahedron inversion.
>
>
> **W2: Surface extraction from Tetspheres**
>
> We extract the surface triangles from each TetSphere to generate surface spheres, and then perform a union of all these surface spheres to obtain the final surface mesh. While there is no theoretical guarantee of manifoldness in the resulting mesh from union operation, our TetSphere optimization is highly regularized to maintain geometric quality. As a result, we did not encounter non-manifold issues in our experiments. We have added this description in Appendix I.
>
> **W3: Topology correctness and analysis on Euler characteristic**
>
> We appreciate your suggestions. However, we did not claim in the paper that our method can guarantee topology correctness. In fact, we acknowledge this as a potential limitation and have discussed it in Sec.6, the Conclusion section. Specifically, the union operation of multiple TetSpheres can result in topology changes that may compromise topology preservation.
>
> Regarding the analysis of the Euler characteristic, it is important to note that many of the ground-truth shapes used in our study are noisy and contain holes that are artifacts of the data, consistent with the dataset used in [DMesh, Son et al., 2024] and evidenced by the rendered results of the ground truth shapes in Fig 5. As such, reporting the differences in Euler characteristics between the reconstructed shapes and the ground-truth shapes would not be informative and could be misleading, as it would reflect the inherent noise in the ground-truth data rather than the performance of our method.
>
> **W4: Mathematical definitions of used metrics**
>
> Thanks for the suggestion. We have included a section in Appendix C providing mathematic definitions of ALR, MR, and CC Diff used across the paper.
>
> **W5: Metrics on mesh quality**
>
> We have added a comparison on the number of triangles in Table 5 of the revised version. Although our method does not have the smallest number of triangles, it achieves the highest mesh quality among all methods. Specifically, our method outperforms NIE and 2DGS, which have approximately twice and four times the number of triangles, respectively, demonstrating superior mesh quality despite the sparser representation.
>
> Regarding the Edge Chamfer Distance, we already included this metric in Table 5 of our original submission, where our method achieved the best result among the compared methods. Additionally, we noted that the TriangleQ metric used in CWF [Xu et al. 2024] is identical to the ALR metric used in our paper. We have added references to this in Appendix C for clarity.
>
> **W6: Intersection between Tetspheres**
>
> The TetSpheres in our optimization are indeed independent branches. Whether a unioned shape is necessary depends on the downstream application task. For rendering tasks, obtaining a unioned shape is not required. However, when a unioned shape is needed, we resolve intersections between TetSpheres by performing a mesh boolean operation, as detailed in Appendix I.
>
> Our silhouette coverage algorithm ensures that the initial TetSpheres are overlapping with each other. While our method does not explicitly guarantee that the final mesh must be a fully connected one, the approach has proven effective in practice: In our experiments, the final mesh obtained after the union operation typically results in a connected shape. Ensuring a strong, theoretical guarantee of mesh connectivity could be an interesting direction for future work.
>
> **W7: Citation problem**
>
> Thank you for pointing out. We have corrected them.
>
>
> **Q1: Topological and manifold guarantees and code.**
>
> Thank you for highlighting these important points. As noted in our responses to Weaknesses, we did not claim in the original paper that our method provides strong topological and manifold guarantees. In fact, we acknowledged these as limitations and potential areas for future research.
>
> We plan to publish the codebase upon acceptance. We hope our responses help clarify these points and assist you in increasing your rating.

---

> > ### Comment · Reviewer_fnQo · 2024-11-22
> >
> > Thanks to the author for the detailed reply. The newly added metric details make it more clear. I still wonder how to extract the 'surface' of the tetrahedron. Is it to extract all four faces of the tetrahedron? Or use some method to judge only the triangles facing outward. Topology and manifold guarantee are also interesting future work. I have improved my rating.

---

> ### Author Response · Authors · 2024-11-22
>
> Thank you for the positive feedback! For surface extraction from TetSpheres, we compute the number of occurrences of each tetrahedron's four faces. Faces that occur twice are interior faces, while those that appear only once are surface faces. Here is the pseudocode reformulated in a numpy-like style for clarity:
>
> ```
> def extract_surface_vertices_and_faces(tetrahedron_faces):
>     """
>     Extract surface vertices and surface faces from a tetrahedral mesh.
>
>     Parameters:
>     tetrahedron_faces: Tx4 numpy array
>         A matrix where each row represents the vertex IDs of a tetrahedron.
>
>     Returns:
>     surface_vertices: numpy array
>         A sorted array of unique vertex IDs that lie on the surface.
>     surface_faces: numpy array
>         A matrix of surface face vertex IDs, remapped to a contiguous range.
>     """
>     # Step 1: Generate all triangular faces of tetrahedra
>     triangular_faces = stack_rows(
>         tetrahedron_faces[:, [1, 2, 3]],
>         tetrahedron_faces[:, [0, 3, 2]],
>         tetrahedron_faces[:, [0, 1, 3]],
>         tetrahedron_faces[:, [0, 2, 1]]
>     )
>
>     # Step 2: Sort vertex IDs in each triangular face to make ordering consistent
>     sorted_triangles = sort_rows(triangular_faces)
>
>     # Step 3: Identify unique faces and count their occurrences
>     unique_faces, face_indices, face_counts = unique_rows(sorted_triangles, return_indices=True, return_counts=True)
>
>     # Step 4: Select faces that occur only once (surface faces)
>     surface_face_mask = (face_counts == 1)
>     surface_faces = unique_faces[surface_face_mask]
>
>     # Step 5: Extract unique vertex IDs from surface faces
>     surface_vertices = unique_elements(surface_faces)
>
>     # Step 6: Map surface vertex IDs to a contiguous range
>     vertex_id_mapping = create_mapping(surface_vertices)
>     remapped_surface_faces = map_indices(surface_faces, vertex_id_mapping)
>
>     # Step 7: Return surface vertices and remapped surface faces
>     return surface_vertices, remapped_surface_faces
> ```

---

### Official Review · Reviewer_u1pH · 2024-10-31

**Soundness:** 4
**Presentation:** 3
**Contribution:** 3
**Rating:** 8
**Confidence:** 4

**Summary:**

This paper presents a Lagrangian geometry representation based on TetSpheres, volumetric tetrahedral spheres that deform to fit the desired geometry. Key applications of TetSphere splatting are demonstrated in monocular and multiview reconstruction. The deformation of TetSpheres is formulated as an energy optimization problem with geometric constraints that prevent the generation of irregular surfaces. Experiments are proposed to compare TetSpheres with state-of-the-art methods based on both Eulerian and Lagrangian geometry representations.

**Strengths:**

The paper is generally well-written, conveying complex concepts and methods in a concise style.

The state-of-the-art is clearly explained and appears to be up-to-date.

The proposed primitive is original and effectively addresses common issues in other Lagrangian geometry representations, such as irregular triangles, non-manifoldness, and floating artifacts.

The energy optimization process for splatting is well-designed and effective, with a convincing initialization algorithm.

The accuracy in monocular and multiview reconstruction applications is competitive with current methods while achieving higher surface quality. This approach is also lighter than other Lagrangian representations in terms of memory and computational complexity.

**Weaknesses:**

The paper proposes a new Lagrangian primitive that is notably more complex than the primitives used in previous methods, such as 3D Gaussians or triangles. Each TetSphere consists of a collection of N tetrahedra, with an apparent trade-off between N and the total number of TetSpheres needed to represent the surface. This trade-off seems to be overlooked in the experimental section, as specific values for the number of TetSpheres and tetrahedra in each experiment are not provided.

The surface is represented as a union of TetSpheres. Due to the proposed splatting optimization, TetSpheres may intersect, leading to artifacts at the intersection points. However, the paper does not specify how intersections between tetrahedra are managed.

**Questions:**

The proposed optimisation does not appear to prevent non-uniform deformation of TetSpheres. However, the resulting reconstruction Aspect Loss Ratio (ALR) is higher than in other methods, suggesting that the tetrahedra within each TetSphere largely preserve their original volume. Is there any factor specifically contributing to the high ALR?


What are the values of N (number of tetrahedra) and M (number of TetSpheres) used in the experiments? How does the trade-off between these values affect reconstruction quality and computational complexity?

More details on the image-to-3D and text-to-3D generation pipelines are needed to ensure the experiments can be reproduced.
Figures 8 to 10 would benefit from captions with additional information about the experimental setup.

---

> ### Author Response · Authors · 2024-11-19
> **Rebuttal**
>
> **W1 & Q2: Trade-off between the number of tetrahedra and the number of TetSpheres**
>
> The number of TetSpheres in our method is determined by the initialization algorithm described in Sec. 4 and Appendix G, controlled by the scaling and offset parameters $\alpha$ and $\beta$ that define the initial radius of each TetSphere. Intuitively, larger values of $\alpha$ and $\beta$ result in a sparser distribution of TetSpheres. We use alpha = 1.2, and beta = 0.07, which result in an average of M = 213 TetSpheres for the shapes used in multi-view reconstruction.
>
> In the revised version, we have added two ablation studies: one analyzing reconstruction performance with respect to different numbers of TetSpheres (Appendix B.1) and another examining the number of tetrahedra per TetSphere (Appendix B.2). When the number of tetrahedra per TetSphere is fixed and the total number of TetSpheres varies, we observe that increasing the number of TetSpheres improves both reconstruction accuracy (as measured by Chamfer distance and Vol. IoU) and surface quality (ALR). However, beyond a certain threshold, these metrics show minimal further improvement. Conversely, when the number of TetSpheres is fixed and the number of tetrahedra per TetSphere increases, reconstruction accuracy does improve, but the gains are less significant compared to increasing the number of TetSpheres. For a more detailed discussion and results, please refer to Fig. 9 and Fig. 10.
>
> **W2: Intersections between tetrahedra**
>
> Whether a unioned shape is necessary depends on the downstream application task. For rendering tasks, obtaining a unioned shape is not required. However, when a unioned shape is needed, we resolve intersections between TetSpheres by performing a mesh boolean operation, as discussed in Appendix I of the revised version. We note that this operation affects a minimal number of triangles relative to the total surface, and we observe little difference in overall mesh quality after performing the union, as intersections occur in only a limited portion of the surface.
>
> **Q1: The proposed optimisation does not appear to prevent non-uniform deformation of TetSpheres. However, the resulting reconstruction ALR is higher than in other methods, suggesting that the tetrahedra within each TetSphere largely preserve their original volume. Is there any factor specifically contributing to the high ALR?**
>
> The ALR is determined by calculating the average ratio of a triangle’s area to its perimeter, with a higher ALR indicating that the triangles are more regular and closer to being equilateral. Initially, the triangles on the TetSphere are isotropic, meaning they are already near-equilateral. Our regularization terms are designed to penalize any nonsmooth deformation gradients of the tetrahedra. Consequently, this approach inherently maintains the isotropy of the triangles to a significant extent.
>
> **Q3: More details on the image-to-3D and text-to-3D generation**
>
> We have added captions with additional information about the experimental setup to the figures in the revised version.

---

> > ### Comment · Reviewer_u1pH · 2024-11-26
> > **Thanks for the reply**
> >
> > I thank the authors for answering my questions and adding new experiments in the paper. It's more clear now and the paper has improved.

---

### Official Review · Reviewer_zZTV · 2024-11-02

**Soundness:** 3
**Presentation:** 3
**Contribution:** 3
**Rating:** 8
**Confidence:** 2

**Summary:**

The manuscript introduces a framework named TetSphere Splatting, which can be used to reconstruct 3D meshes from multi-view images. TetSphere uses tetrahedron spheres to represent an object. During optimization, deformation field is predicted to relocate vertex points to minimize cost. Experiments in multi-view reconstruction, image/text to 3D shows potential applications of TetSphere Splatting.

**Strengths:**

-	The problem is of interest to the research community.
-	The formulation of TetSphere Splatting is novel, and an interesting theoretical perspective from prior work.
-	The experiments setup is diverse, including many applications.

**Weaknesses:**

Overall, the paper is quite interesting. However, my concern about the paper is its claim, where experimental results cannot back up.

-	TetSphere is claimed to have superior surface quality due to the fact Marching Cubes is not needed. However, all qualitative results seem to suggest that the TetSphere needs to be sufficiently dense to produce detailed structures. This leads to unnecessarily dense triangles, which can also be achieved by very dense marching cubes resolution. What is the advantage in this case?

-	Moreover, while the manifoldness of TetSphere is guaranteed, the baselines compared are all Eulerian representations without sufficient regularization. Approaches such as NeuS and VolSDF use volume rendering to achieve manifold meshes. While I understand the faster optimization process aspect, can authors comment on TetSphere’s advantage over NeuS/VolSDF from the quality perspective?

-	Lastly, the claimed superior surface quality is not backed up by smaller Chamfer Distance, such as in Table 2. Can authors provide more explanation?

**Questions:**

-	When supervised purely based on depth/mask, the interior TetSphere not visible to any images will not receive gradients, except for regularization. Is this the main reason manifoldness is maintained?

-	Mosaic SDF [a] is one missing work that also follows the Lagrangian framework, where a set of volumes are moved in space.

[a] Yariv, L., Puny, O., Gafni, O., & Lipman, Y. (2024). Mosaic-sdf for 3d generative models. In Proceedings of the IEEE/CVF Conference on Computer Vision and Pattern Recognition (pp. 4630-4639).

-	Clarification needed, below variables are not properly defined:
 - L265: what is N?
 - L266: what is T?
 - L309: what is n?

---

> ### Author Response · Authors · 2024-11-19
> **Rebuttal**
>
> **W1: Density of TetSpheres and advantanges over dense marching cubes.**
>
> Our method employs a Lagrangian representation, whereas the Marching Cubes algorithm is typically used with an Eulerian framework. Consequently, each method retains the inherent characteristics of its respective representations, which we have detailed in the introduction. One notable advantage of using a Lagrangian representation is its ability to represent a shape with fewer parameters compared to its Eulerian counterpart.
>
> Regarding the density of TetSpheres, it is not inherently necessary for achieving high surface quality. In the revised version, we have included two additional experiments to support this claim: (a) a comparison of the average number of triangles in the reconstructed shapes, as shown in Table 5, where our method, while not producing the highest triangle count, still achieves superior mesh quality; (b) an ablation study comparing the impact of the number of TetSpheres vs. the number of tetrahedra within each TetSphere, as detailed in Appendix B.2 and Figure 10 (b). While increasing the number of tetrahedra within each TetSphere can lead to performance improvements, these gains are relatively minor compared to increasing the overall number of TetSpheres, demonstrating that effective coverage is more crucial than the density of each TetSphere.
>
> **W2: TetSphere’s advantage over NeuS/VolSDF from the quality perspective**
>
> Besides manifoldness and faster optimization speed, TetSphere also outperforms Eulerian methods such as NeuS/VolSDF in terms of the regularity of surface triangles. This is demonstrated by the single-view reconstruction results presented in Table 2, where the baseline methods SyncDreamer and Wonder3D, which use NeuS as their geometry representation, are compared. Our approach outperforms these methods in terms of the Area-Length Ratio (ALR), indicating more regular and well-formed surface triangles and, consequently, superior triangle quality.
>
> **W3: Explanation of chamfer distance**
>
> In our paper, we specifically refer to mesh quality using the three metrics of ALR, MR, and CC Diff which assess triangle regularity, manifoldness, and structural integrity and coherence. This is consistent with prior work such as FlexiCubes [Shen et al., 2023b] and DMesh [Son et al., 2024]. Chamfer distance, on the other hand, is used as a metric for reconstruction accuracy rather than surface quality, as it measures the distance between sampled points on the reconstructed mesh and the ground truth mesh, without evaluating the regularity or quality of the surface itself. While our method may not achieve the lowest chamfer distance, it still delivers competitive reconstruction accuracy, as evidenced by the Volume IoU metric. Additionally, our method achieves significantly superior performance in terms of mesh quality as reflected in the ALR, MR, and CC Diff metrics.
>
> **Q1: Gradient on invisible TetSphere interior and manifoldness**
>
> The interior TetSphere indeed only receives gradients from regularization terms. The volumetric regularization helps guide the internal structure to remain consistent and avoid non-manifold artifacts.
>
> **Q2: Missing work of Mosaic SDF**
>
> We have added a discussion on Mosaic SDF in the Related Work section. Specifically, Mosaic SDF is designed for 3D generation tasks where ground-truth shapes are provided, as it requires a surface input. In contrast, our method only requires multi-view images as input, enabling it to support reconstruction tasks without the need for pre-existing surface information.
>
> **Q3: Clarification needed on the definition of variables**
>
> In the revised version, we have added explanations for the variables used: N represents the number of vertices for each TetSphere, and T denotes the number of tetrahedra within each TetSphere. The variable n on line 309 was a typo and has been corrected to M. We appreciate your attention to detail.

---

> > ### Comment · Reviewer_zZTV · 2024-11-22
> >
> > Hi authors,
> >
> > Thank you very much for the detailed reply. These are very helpful. I have the following questions based on the reply.
> >
> > W1: The results make sense to me. However, I wonder if it is possible that TetSpheres can be optimized w.r.t to other inputs such as edgeloops. Given a high coverage using many TetSpheres initially, would it possible to gradually reduce/merge the vertices?
> >
> > W2: While the result in Table 2 shows promising results, the setting is quite ill-posed for NeuS/VolSDF. The result highly depends on the consistency of multi-view images. Moreover, the sparse view setting is challenging inherently for these methods that do not assume any priors. Showing results in the setting in Table 1 would be much more convincing. Is it possible to have such comparison? Also how many views are there in the multi-view setting?

---

> > > ### Author Response · Authors · 2024-11-25
> > >
> > > Thanks for your feedback.
> > >
> > > W1.1: TetSpheres can be optimized with respect to edge loops, which we leave for future work.
> > >
> > > W1.2: While merging vertices or performing adaptive remeshing during optimization is technically feasible for tasks like multi-view reconstruction, it increases running time complexity and disrupts consistent texture parameterization. This consistency is essential for generative tasks such as text-to-3D, where the texture image itself is an optimization variable (see Appendix H).
> > >
> > > W2: We added results for NeuS and VolSDF to Table 1. For DMesh and FlexiCubes, we used their original codebases, which randomly sample cameras per iteration, while other methods used 120 views in the multi-view setting. Both NeuS and VolSDF, as Eulerian methods, are computationally slow (~4 hours per optimization on an A100 GPU). Our method achieves better mesh quality and is computationally efficient.

---

> ### Comment · Reviewer_zZTV · 2024-11-25
>
> Thanks for the additional results. I have increased my rating given the clarifications from the authors.
>
> W1.1: adding such a discussion could be interesting! I have not seen any work able to achieve this yet.
>
> W2: This sufficiently addressed my concerns.

---

### Official Review · Reviewer_hHtw · 2024-11-03

**Soundness:** 3
**Presentation:** 4
**Contribution:** 3
**Rating:** 8
**Confidence:** 2

**Summary:**

The paper proposes a shape reconstruction method that takes a set of images as input and fits a 3D model with reflectance properties to them. In line with a large body of recent work, it combines a rendering loss with shape regularization to directly optimize a 3D scene representation by "simple" numerical descent. The problem is particularly challenging as the paper uses a "Lagrangian" representation, specifically, a tetrahedral mesh that co-moves with the surface points obtained.

As recent work, such as [Nicolet 2021] has pointed out, the success of such an approach critically hinges upon suitable regularization, as the rendering loss is ambiguous and it is easy to get stuck in a bad local minimum with nonsensical mesh structures (which has been an issue for a long time in attempting such direct shape optimization against relatively weak data constraints).

The key contribution of this paper is to use a volumetric representation, a tet-mesh, with a penalty against inversion of elements along with an incrementally applied smoothness regularizer (a bi-harmonic energy applied to deformation gradients of the current mesh) that promotes changes in low-frequency shape first.

The paper provides very convincing results and shows a number of interesting applications, some of which only made it into the appendix.

**Strengths:**

I think that the main strength of the paper is the idea itself: Regularization is important to avoid bad mesh structure, and a volumetric representation intuitively gives much more leeway to regularization over a pure thin-shell / surface model (the inductive bias of this being a solid is stronger than regularizing details on thin surface). The proposed setup with inversion constraints and smoothness looks plausible and seems to work very well in practice. I should add that I am not actively working in this area, so I might miss some related work; but if this kind of approach has not yet been tried, the approach itself seems worth publishing.

The results are also convincing in terms of quality and versatility. The paper is very well written, nicely illustrated and enjoyable to read. Comparisons against some related recent method also underline the quality of the results quantitatively.

**Weaknesses:**

In my opinion, the main downside is that the paper is rather light on analysis: Why does it use the bi-harmonic energy on deformation gradients? What does this actually mean (application to derivatives should create higher-order smoothness, or in terms of a Fourier-perspective, add an additional high-pass filter in front of the regularizer)? What happens, if other regularizers are used? How important is the non-inversion term, and how brittle are parameter choices? It would also be interesting to discuss the relation to other attempts at stricter regularization, such as Nilcolet et al.'s work. I would guess that the volumetric approach is strictly superior, if fully exploited; showing something like this experimentally would make the paper much stronger.

Some of the limitations could also been discussed more clearly. For example, the paper only produces piecewise meshes, which are overlayed; it does not solve the problem of globally fitting a single tet-mesh. Comparisons in mesh quality should take this into account. Further, this also shows implicitly that the method is still to brittle to handle strong deformations during optimization; otherwise, one could just fit a single sphere globally. In this context, as well as in general, it would be instructive to show how to break the method by slowly stepping outside its "convergence radius" where good results can be obtained into cases where results are unsatisfactory. My criticism here is not that there are limitations but that the paper could be strengthened by studying them more in depth.

Finally, the paper seems like a better fit to a conference on computer vision or graphics in terms of methods and problems, but I would consider the topic close enough not to exclude it.

**Questions:**

When does the method break? At which point does the proposed machinery fail to fit a reasonable geometry to data, and how do the problems look like / introduce themselves? Which conceptual or technical challenges would be needed to overcome to fix this?

Does the method have fundamental advantages over previous representations? If so, can you make a formal or experimental argument to convince the reader? If the differences are more nuanced, what are the key trade-offs one has to make here?

---

> ### Author Response · Authors · 2024-11-19
> **Rebuttal (1/2)**
>
> **W1.1: The use of bi-harmonic energy on deformation gradients**
>
> The incorporation of biharmonic energy in our method is inspired by its proven efficacy in geometry processing, as highlighted by prior research [1]. Furthermore, penalizing the nonsmoothness of transformations (or deformation gradient) across a domain is a well-explored area that has demonstrated its utility in various geometry processing tasks such as shape deformation [3], deformation transfer [2], and mesh alignment [4].
>
> For the surface regions where observations are available, the surface deformation is primarily influenced by the rendering loss. Conversely, in regions lacking direct observations, our regularization term plays an important role by penalizing high-frequency deformations resulting from nonsmooth deformation gradients. This approach ensures reliable results.
>
> The biharmonic energy offers stronger regularization compared to first-order harmonic energy, as demonstrated in [1]. Although theoretically higher-order smoothness energies can be considered, in practice, they often lead to numerical instability, as discussed in [5]. Therefore, biharmonic energy strikes an balance by providing effective regularization while avoiding numerical issues.
>
>
> [1] On Linear Variational Surface Deformation Methods, Botsch et al., 2008
>
> [2] Deformation Transfer for Triangle Meshes, Sumner et al., 2004
>
> [3] Modeling of Personalized Anatomy using Plastic Strains, Wang et al., 2021
>
> [4] Optimal Step Nonrigid ICP Algorithms for Surface Registration, Amberg et al., 2007
>
> [5] Libigl, https://libigl.github.io/
>
> **W1.2: How important is the non-inversion term, and how brittle are parameter choices?**
>
> The non-inversion term plays a critical role by preventing surface flipping and overlapping. In Section 5.4 and Fig. 6, we analyze the effects of energy coefficients and illustrate how changes to these coefficients affect reconstruction results using an Armadillo shape. We found that larger coefficients yield smoother surfaces, whereas using excessively small coefficients or omitting the non-inversion regularization can result in tetrahedron inversion, ultimately causing artifacts on the surface.
>
> **W1.3: Experimental comparison with [Nicholet et al. 2021]**
>
> We have added a comparison with [Nicholet et al. 2021] in Appendix A in the revised version. Table 4 and Fig. 8 show that our method outperforms [Nicholet et al. 2021], particularly for shapes with complex topologies. Unlike these single-sphere, surface-regularized approaches, which struggle with objects featuring multiple holes (e.g., the dress and sorter), our method, using multiple TetSpheres and volumetric regularization, preserves thin regions and open areas without unrealistic closures. Additionally, our volumetric approach effectively prevents artifacts like folded, overlapping surfaces and crumpled regions seen in [Nicholet et al. 2021], highlighting the benefits of a volumetric method for achieving high-quality reconstructions.
>
> For a more throughtout discussion about [Nicholet et al. 2021], please refer to [[AA94] W1](https://openreview.net/forum?id=8enWnd6Gp3&noteId=mo1geI0stA).
>
> **W2: Limitations could be discussed more clearly.**
>
> We have added a section in Appendix K regarding the overlaid piecewise meshes rather than fitting a single tetrahedra mesh. We have also added a paragraph in Appendix I describing how to obtain a unioned shape from TetSpheres. We note that this operation affects a minimal number of triangles relative to the total surface, and we observe little difference in overall mesh quality after performing the union, as intersections occur in only a limited portion of the surface.
>
> **W3: More studies into cases where results are unsatisfactory.**
>
> In the revised version, we have added multiple ablation studies (Fig. 8, 9, and 10). Regarding the comment on fitting a single sphere globally, the ablation study on the number of TetSpheres used for reconstruction, detailed in Appendix B.1, explores how the number of TetSpheres affects reconstruction performance, with qualitative results shown in Fig. 9.
>
> When an appropriate number of TetSpheres is used, our method achieves the best results. Cases where the number of TetSpheres is too small relative to the complexity of the shape – such as attempting to use only a single global sphere for highly detailed models like those shown in Fig. 9 – may result in unsatisfactory reconstructions.
>
> We hope these additions and clarifications address your concerns and more clearly highlight the strengths and limitations of our approach.

---

> > ### Comment · Reviewer_hHtw · 2024-11-21
> >
> > Dear authors, thanks for the detailed feedback! I think that the additional experiments strengthen the paper further (also thanks for pointing out the comparison within Fig. 6, which I had initially overlooked).
> >
> > My remaining main suggestions for a potential further revision would be textual in the sense of addressing limitations and unknowns more aggressively in the main text (maybe pointing to an appendix for all details). I would also think that a including detailed motivation of the regularizer (W1.1; maybe appendix) might help the reader understand the choices.

---

> > > ### Author Response · Authors · 2024-11-21
> > >
> > > Thanks for the suggestions! We have just updated the submission:
> > >
> > > 1. A sub-section in the Conclusion discussing the limitations, which points to Appendix K.
> > >
> > > 2. A detailed discussion on the motivation of the regularization terms in Appendix L (referenced in main text l264)

---

> ### Author Response · Authors · 2024-11-19
> **Rebuttal (2/2)**
>
> **Q1: Potential failure cases and conceptual or technical challenges**
>
> As mentioned in our response to the weaknesses, our method requires a sufficient number of TetSpheres to ensure that the output shape is of high quality. This necessitates a close match between the resolution of the initial TetSphere distribution and the level of detail required by the target geometry. Failure cases can arise in reconstruction tasks when the number of input views is extremely sparse, when the views do not sufficiently cover the target object (e.g., missing key angles or areas), or when the views lack detail due to low resolution or poor alignment. In such scenarios, the number of TetSpheres may be insufficient to faithfully cover the entire target shape, leading to incomplete or lower-quality reconstructions.
>
> **Q2: Key trade-offs over previous representations.**
>
> Our method can be viewed as balancing the trade-off between the number of primitives used for shape representation and the complexity of each individual primitive. In general, using fewer primitives requires each primitive to be more complex to adequately capture the shape, necessitating intricate regularization and processing schemes. For example, [Nicolet 2021] employs a single surface sphere with additional remeshing steps to manage large deformations and maintain mesh quality. On the other hand, using simpler primitives, as seen in methods like DMesh (which relies on surface triangles) and Gaussian Splatting (GS, which uses point clouds), requires a large number of primitives to capture intricate shapes. While this approach reduces the need for complex regularization within each primitive, it sacrifices overall mesh quality due to limited interactions between the primitives.
>
> Our method, leveraging structured TetSpheres, strikes a balance between these extremes. By using multiple simple yet structured volumetric primitives, we enable strong regularization within each TetSphere without necessitating complex deformation schemes or intermediate remeshing. This allows for more robust and high-quality reconstructions, maintaining a balance between computational efficiency and reconstruction fidelity.

---

### Official Review · Reviewer_AA94 · 2024-11-03

**Soundness:** 3
**Presentation:** 3
**Contribution:** 2
**Rating:** 6
**Confidence:** 5

**Summary:**

This paper proposes a method for reconstructing 3D geometry from multiview images. The geometry is represented by Tetrahedral meshes of sphere topology that are optimized in standard fashion w.r.t a rendering loss, along with two geometric regularizes - the biharmonic energy to ensure a smooth deformation, and a barrier term that prevents tetrahedra from inverting.

**Strengths:**

The paper revisits the idea of using meshes for 3D reconstruction, and reaffirms that an explicit representation is extremely efficient in producing high quality 3D geometry from images. The paper is written in a clear way and is easy to understand.

**Weaknesses:**

1. My main concern with the paper is its novelty and contribution: the paper essentially proposes to deform a mesh w.r.t a visual loss which is standard (e.g., [Nicloet et al. 2021]), while using two standard regularizes to guarantee a "good" deformation (smooth and without inversions). This is a very standard approach. The only part that seems less explored is the use of tetrahedra instead of triangles, however this of course has been explored extensively outside of the context of differentiable rendering, hence the only part I can deem truly novel is "using tetrahedra instead of triangles in tandem with multiview reconstruction". This feels like a marginal contribution. It could be argued to be a very practical approach which may convince me to champion it, however that leads me to my second concern:
2. Evaluation seems somewhat selective, and the methods compared to do not strike me as the immediate alternatives. Namely, I find [Nicolet et al. 2021] as a main contender, as well as "Continuous remeshing for inverse rendering" which is uncited. Many other techniques that cite [Nicolet et al. 2021] can be considered.
3. The argumentation for the use of a volumetric mesh instead of a triangle mesh needs to be justified further through experiments. It seems the only argument for a full discretization of the *volume* (as opposed to the surface via a triangle mesh), is to regularize the volumetric deformation with the two losses. This seems like a significant overkill, as simpler regularizers could be employed on the triangle mesh (again, e.g., Nicolet et al. 2021) to ensure it behaves well.

**Questions:**

- can you explain why did you choose to not compare to [Nicolet et al. 2021] and methods that adopted it later?
- you do not cite "Continuous remeshing for inverse rendering" - how do you think your method would fair in respect to it?
- are there other advantages to using a tetrahedral mesh as opposed to triangle, except the two regularization losses?

---

> ### Author Response · Authors · 2024-11-19
> **Rebuttal (1/2)**
>
> **W1: Novelty and contribution compared to work such as [Nicolet et al. 2021] and [Palfinger 2022]**
>
> Thank you for highlighting the relevance of [Nicolet et al. 2021]. In addition to the volumetric nature of our TetSphere, a key distinction of our approach compared to [Nicolet et al. 2021] and its subsequent works is the implementation of **multiple tetrahedral spheres** instead of just one. In the multi-view reconstruction examples presented in our paper, the average number of TetSpheres used is approximately 210. This significantly alters the complexity of the fitting process and introduces new methodological advantages:
>
> Both [Nicolet et al. 2021] and our approach employ regularizations to prevent extreme deformations that can compromise geometric quality, thereby inherently limiting each sphere's expressivity to a certain extent. [Nicolet et al. 2021] address this limitation by using intermediate remeshing to accommodate topological changes and manage drastic deformations when the target shape significantly differs from the sphere. However, remeshing results in the reparameterization of the texture, which poses challenges in the context of 3D shape generation pipelines. Maintaining a consistent texture parameterization throughout the optimization process is crucial, as the texture image itself is an optimization variable (see Appendix H). Additionally, remeshing can introduce undesired meshing complications when the surface undergoes topological changes. To prevent topological changes during each iteration, a single primitive must be initialized with a topology that matches the target, as demonstrated in Fig. 7 of the original paper.
>
> By contrast, our method eliminates the need for intermediate remeshing by utilizing multiple TetSpheres alongside stronger regularization terms, including biharmonic energy and non-inversion constraints. By leveraging multiple TetSpheres, each TetSphere undergo a smaller deformations, ensuring robust shape reconstruction. This approach also adeptly reconstructs complex shapes with numerous holes, as demonstrated in the sorter and dress examples in Fig. 8.
>
> Even with the use of multiple spheres, our method achieves comparable wall-clock time with [Nicolet et al. 2021] and [Palfinger 2022], as we parallelize the optimization process for all TetSpheres. On an A100 GPU, the average running time is approximately 4 minutes.
>
> **W2: Evaluations compared to [Nicolet et al. 2021] and "Continuous remeshing for inverse rendering [Palfinger 2022]"**
>
> In the revised version, we show both quantitative and qualitative comparisons with [Nicolet et al. 2021] and [Palfinger 2022] in Table 4 and Fig. 8 in Appendix A. We used their publicly available codebases and performed multi-view reconstruction on our dataset as described in Sec. 5.2.
>
> Our method outperforms these approaches, particularly for shapes with complex topologies. Both [Nicolet et al. 2021] and [Palfinger 2022] use a single sphere for reconstruction, which limits their ability to handle objects with multiple holes, such as the dress and the sorter. For the dress example (the second and the third columns in Fig. 8), both baseline methods exhibit unrealistic closure of the open areas. In contrast, our approach accurately represents the thin regions and preserves the open areas due to the use of multiple TetSpheres. For the sorter example (the fourth and the fifth columns), the results from [Nicolet et al. 2021] exhibit folded, overlapping surfaces with noticeable crumpled regions. This is due to their surface-based deformation regularization, which does not prevent the interior volume from shrinking to zero. On the other hand, our tetrahedral-based method, with its volumetric regularization, mitigates these issues.
>
> **W3: The argumentation for the use of a volumetric mesh instead of a triangle mesh needs to be justified further through experiments.**
>
> Incorporating a volumetric mesh along with volumetric regularization terms is both essential and well-founded. Regularization approaches that concentrate only on surface deformation (e.g. [Nicolet et al. 2021]) can result in undesired folded and overlapping surfaces because they neglect to address or regularize the interior volume. This issue is illustrated in Fig. 8. On the other hand, volumetric regularization intrinsically reduces these artifacts by penalizing nonsmoothness of the volumetric deformation gradient and preventing volume inversion. The advantages of this approach are demonstrated in our results.

---

> > ### Author Response · Authors · 2024-11-19
> > **Rebuttal (2/2)**
> >
> > **Q1: Can you explain why did you choose to not compare to [Nicolet et al. 2021] and methods that adopted it later?**
> >
> > Initially, our hypothesis is that our method using multiple primitives had advantages over those using a single primitive (e.g., [Nicolet et al. 2021]). This led us to compare our method with Lagrangian approaches like DMesh and Gaussian Splatting, which employ significantly more primitives.
> >
> > However, we acknowledge the importance of comparing our method with single-primitive approaches, such as [Nicolet et al. 2021] and [Palfinger et al. 2022]. In response, we have included results in the revised version that directly compare our method to these single-primitive methods. We hope this addition offers a more comprehensive experimental evaluation.
> >
> > **Q2: You do not cite "Continuous remeshing for inverse rendering" - how do you think your method would fair in respect to it?**
> >
> > Both citations and experimental comparisons have been added in the revised version. Additionally, we have included a paragraph in the related work section discussing these inverse rendering methods.
> >
> > **Q3: Are there other advantages to using a tetrahedral mesh as opposed to triangle, except the two regularization losses?**
> >
> > Additional advantages to using tetrahedral meshes include: Their volumetric nature allows for stronger regularization and more robust optimization processes, which reduce artifacts like surface folding. This volumetric consistency is also beneficial for downstream applications, such as simulations, finite element analysis, and volume rendering, where a reliable volume representation is essential.

---

> > ### Comment · Reviewer_AA94 · 2024-11-22
> >
> > Thank you for your clarifications. I appreciate the empirical practicality of the method, hence will raise my score. However, I cannot really champion the paper, as I still think that for the most part it uses well-known, existing building blocks, for a straightforward approach ("replace Gaussians with tet meshes). If you think there's a unique technical novelty aside from the proposal to use well-known regularizers for the deformation, along with simply replacing GS with tet meshes, please correct me.

---

> > > ### Author Response · Authors · 2024-11-25
> > >
> > > Thank you for your thoughtful feedback and for raising your score. We sincerely appreciate your acknowledgment of the method’s empirical practicality and understand your perspective regarding its technical novelty. While our approach builds upon established concepts, we believe that integrating these components into TetSphere splatting as a novel representation provides greater flexibility and robustness for handling complex geometries. Additionally, the combination of volumetric regularization with tailored initialization and optimization effectively addresses the limitations of previous methods, such as Gaussian Splatting, in producing high-quality reconstructions. We hope this clarifies the unique technical contributions of our work and inspires further advancements in this area.

---

### Author Response · Authors · 2024-11-19
**General Response**

We sincerely appreciate the detailed reviews and the thoughtful feedback provided by all reviewers and the area chair. In addition to addressing specific comments from each reviewer, we would like to outline our primary contributions.

* **[Idea]** The proposal of a Lagrangian volumetric representation with regularization was highlighted as intriguing and original [hHtw]. The use of tetrahedrons for shape representation was noted as a valuable and effective approach [fnQo].
* **[Methodology]** The formulation of TetSphere Splatting was recognized as novel, offering an interesting theoretical perspective on prior work [zZTV]. The energy optimization process was praised for being well-designed and supported by a convincing initialization algorithm [u1pH].
* **[Experiments]** Our experiments in both multi-view and single-view reconstruction tasks demonstrated superior mesh quality and competitive reconstruction accuracy. Reviewers noted that the experimental setup was diverse [u1pH], the proposed method surpasses or catches up with SOTA [fnQo], and results are convincing [hHtw].
* **[Presentation]** The manuscript was commended for its clarity and well-written quality [AA94, hHtw, u1pH].

During the rebuttal period, we have made the following revisions to the manuscript as recommended by the reviewers, as highlighted in red in the uploaded PDF:
* Experimental comparisons with inverse rendering approaches [Nicolet et al., 2021] and [Palfinger 2022] (Appendix A, Figure 8, and Table 4).
* An ablation study on the number of TetSpheres (Appendix B.1, Fig. 9, and Fig. 10(a)).
* An ablation study on the number of tetrahedra per TetSphere (Appendix B.2, Fig. 10(b)).
* Mathematical definitions of the metrics used to evaluate mesh quality (Appendix C).
* A simple experimental ablation on tetrahedron inversion  (Appendix D).
* Additional metrics related to the number of triangles (Table 5).
* Explanations on obtaining the final surface mesh (Appendix I).
* Discussed limitations of the method in more detail (Appendix K).
* Incorporated all recommended references, discussions of related work, hyperparameter descriptions, writing improvements, and figure captions.

We hope our responses address all reviewers' concerns and help improve the review scores. We thank all reviewers and the AC again for their time and efforts!

---

### Comment · Area_Chair_q8iW · 2024-11-25
**Last day for interactive discussions!**

Dear authors and reviewers,

The interactive discussion phase will end in one day (November 26). Please read the authors' responses and the reviewers' feedback carefully and exchange your thoughts at your earliest convenience. This would be your last chance to be able to clarify any potential confusion.

Thank you,
ICLR 2025 AC

---

### Meta-Review · Area_Chair_q8iW · 2024-12-19

**Metareview:**

The submission received positive reviews from all the reviewers. The reviewers generally appreciate the clarity, recognize the novelty of the method, and are convinced by the positive experimental results. After reading the paper, the reviewers' comments and the authors' rebuttal, the AC agrees with the decision by the reviewers and recommends acceptance.

**Additional Comments On Reviewer Discussion:**

The reviewers raised questions mostly regarding selective or missing comparisons (AA94, hHtw, zZTV, fnQo) and artifacts (u1pH, fnQo). The questions were addressed by the authors in good detail. Reviewers AA94, zZTV and fnQo were convinced by the responses and raised their ratings. The AC agrees with the evaluation.

---

### Decision · Program_Chairs · 2025-01-22

Accept (Oral)